# HYBRID DIRECTIONAL GRAPH NEURAL NETWORK FOR MOLECULES

**Junyi An**[1]*, **Chao Qu**[2]†, **Zhipeng Zhou**[2], **Fenglei Cao**[3], **Yinghui Xu**[4], **Yuan Qi**[4], **Furao Shen**[1]†

[1]State Key Laboratory for Novel Software Technology, Nanjing University
[2]INFLY TECH (Shanghai) Co., Ltd.
[3]Shanghai Academy of Artificial Intelligence for Science
[4]Artificial Intelligence Innovation and Incubation (AI[3]) Institute, Fudan University

## ABSTRACT

Equivariant message passing neural networks have emerged as the prevailing approach for predicting chemical properties of molecules due to their ability to leverage translation and rotation symmetries, resulting in a strong inductive bias. However, the equivariant operations in each layer can impose excessive constraints on the function form and network flexibility. To address these challenges, we introduce a novel network called the Hybrid Directional Graph Neural Network (HDGNN), which effectively combines strictly equivariant operations with learnable modules. We evaluate the performance of HDGNN on the QM9 dataset and the IS2RE dataset of OC20, demonstrating its state-of-the-art performance on several tasks and competitive performance on others. Our code is anonymously released on https://github.com/ajy112/HDGNN.

## 1 INTRODUCTION

In recent years, Graph Neural Networks (GNNs) have experienced remarkable success across various domains, including social networks (Kipf & Welling, 2017), physical systems (Battaglia et al., 2018), computational biology (Townshend et al., 2019; Chami et al., 2019), and many others. Among these domains, the application of GNNs in computational chemistry has garnered significant interest due to its promising ability to predict energy and other chemical properties with comparable accuracy to quantum mechanical simulation methods (such as DFT), while achieving speedups of 4-5 orders of magnitude (Behler & Parrinello, 2007; Gilmer et al., 2017).

Nonetheless, regular GNNs alone are insufficient for accurate modeling of molecules, as they overlook crucial chemical constraints such as invariance and equivariance (Fuchs et al., 2020). For instance, both molecular and atomic energies remain invariant under rotation transformations, whereas forces exhibit equivariance. To guarantee the rotational invariance of a predicted property, Schnet (Schütt et al., 2018) and HIP-NN (Lubbers et al., 2018) restrict the network's inputs to only depending on interatomic distance, while this leads to a loss of directional and equivariant information (Miller et al., 2020). Directional GNNs go a step further by explicitly incorporating bond angles and dihedral angles information in GNNs (Gasteiger et al., 2020; 2021).

To capture deep directional features for both invariant and equivariant interactions, equivariant neural networks have been proposed (Brandstetter et al., 2021; Liao & Smidt, 2023). These networks utilize group representations to construct steerable group convolutions or message passing blocks, ensuring equivariance throughout the entire network. While equivariant neural networks exhibit appropriate inductive biases for molecular systems, the operations under strictly equivariant constraints will limit the expressiveness of the network. This loss of expressiveness impedes the model's ability to effectively learn the intricate interactions between atoms. An analogous situation can be observed in convolutional neural networks, where imposing equivariance results in overly constrained networks and suboptimal performance (Romero & Lohit, 2022). In contrast, relaxing equivariance approaches (Romero & Lohit, 2022; Wang et al., 2022) allow for learning approximate equivariance while capturing the nuances of reality, thereby enabling strong generalization. Recently, Zitnick et al.

---

*e-mail: junyian@smail.nju.edu.cn

†Corresponding authors: Chao Qu is the corresponding author in INFLY TECH, and Furao Shen is the corresponding author in Nanjing University.

(2022) relaxes the equivariance constraint in GNNs and accomplishes better performance by using additional coefficients. This work has demonstrated leading performance on challenging tasks such as OC20 (Chanussot et al., 2021). To further explore more effective molecular model, we investigate the impact of relaxing equivariance based on theories of expressive power (Dym & Maron, 2021; Joshi et al., 2023). It is worth noting that while certain equivariant GNNs with infinite degree of group representation $l$ have been proven to possess universal approximation capabilities for any equivariant function and the expressive power of equivariant GNNs can be enhanced by increasing degree number $l$ during conducting Clebsch-Gordan (CG) tensor product (Dym & Maron, 2021), it is impractical to employ the high-degree CG product due to the substantial computational costs. Given the aforementioned dilemma, we observe that certain specialized operations can achieve the expressive power of high-degree CG product with significantly lower computational complexity, albeit introducing non-equivariance. Detailed introductions can be found in Section 2.4. Building upon our observation, we aim to learn appropriate relaxed equivariant features, which can achieve expressive power of high-degree representation and ensure approximate equivariance.

In this paper, we propose the Hybrid Directional Graph Neural Network (HDGNN), a message-passing GNN designed to enhance expressiveness by relaxing equivariance. To accomplish this, we replace strictly equivariant building blocks, such as the CG tensor product, commonly used in equivariant GNNs (Griffiths & Schroeter, 2018; Kondor et al., 2018), with flexible neural networks. However, naively applying neural networks to construct these modules undermines equivariance and degrades the overall network performance, as evidenced by our ablation study in Section 5.2. To this end, we propose a neural structure that incorporates subtle designs for learning equivariant properties. Furthermore, this structure is combined with strictly equivariant operations through a dynamic module. With this design, HDGNN can achieve an automatic balance between equivariance and expressiveness. If we discard certain operations in our building blocks or replace the learned messages with fixed spherical harmonic representations, our networks may degenerate to the vanilla equivariant GNNs which may subsume the existing works (Kondor et al., 2018; Thomas et al., 2018). We evaluate our HDGNN on the QM9 benchmark (Ramakrishnan et al., 2014) and the IS2RE dataset of OC20 (Chanussot et al., 2021). On the OC20 dataset, HDGNN outperforms state-of-the-art (SOTA) methods and leads a significant improvement on unseen samples. For the QM9 dataset, HDGNN achieves the best results on several tasks and comparable results on the remaining ones. We provide extensive ablation studies on non-equivariant modules and the network structure.

We summarize our main contributions as follows: (i) A neural architecture for learning relaxed equivariant representations, which can enhance the expressive power suffering from the limitations of finite low-degree group representation and ensure approximate equivariance; (ii) The ablation study sheds light on our building blocks and shows their necessities. It may pave the way for the future study on the approximately equivariant neural network.

## 2 Preliminary

In this section, we make a brief review of the necessary mathematical background to depict our model, which includes equivariance, spherical harmonics, the Clebsch-Gordan (CG) tensor product, and so on. Additionally, we briefly describe the universal approximation of equivariant function. More detailed introductions are deferred to Appendix A. We list the notations frequently used in the following. We denote unit sphere as $S^2$, where spherical coordinates $(\theta, \varphi)$ are polar angle and azimuth angle, respectively. The symbol $\mathbb{R}$ stands for the set of real numbers while $\mathbf{R}$ represents the rotation matrix for 3D vectors. We use $\mathbf{G}$ to denote the group and $SO(3)$ to denote the special orthogonal group, i.e., the 3D rotation group. We use $\circ$ and $\otimes$ to represent the element-wise product and the CG tensor product, respectively.

### 2.1 Equivariance and Invariance

Given a group $\mathbf{G}$ and transformation parameter $g \in \mathbf{G}$, a function $\phi : \mathcal{X} \to \mathcal{Y}$ is called equivariant to $g$ if it satisfies:
$$T'(g)[\phi(x)] = \phi(T(g)[x]), \tag{1}$$
where $T'(g) : \mathcal{Y} \to \mathcal{Y}$ and $T(g) : \mathcal{X} \to \mathcal{X}$ denote the corresponding transformations over $\mathcal{Y}$ and $\mathcal{X}$, respectively. Invariance is a special case of equivariance where $T'(g)$ is an identity transformation. It says that the output of $\phi$ is unaffected by the transformation applied to the input. In this paper, we mainly focus on the $SO(3)$ equivariance and invariance, since it is closely related to the interactions between atoms in molecule [1].

---

[1]Invariance of translation is trivially satisfied by taking the relative positions as inputs.

## 2.2 SPHERICAL HARMONICS AND STEERABLE VECTOR

Spherical harmonics, a class of functions defined over the sphere $S^2$, form an orthonormal basis and have some special algebraic properties widely used in equivariant models (Kondor et al., 2018; Cohen et al., 2018). In this paper, we use the real-valued spherical harmonics denoted as $\{Y_m^l : S^2 \rightarrow \mathbb{R}\}$, where $l$ and $m$ denote degree and order, respectively. It is known that any square-integrable function defined over $S^2$ can be expressed in a spherical harmonic basis via

$$f(\theta, \varphi) = \sum_{l=0}^{\infty} \sum_{m=-l}^{l} f_m^l Y_m^l(\theta, \varphi), \tag{2}$$

where $f_m^l$ is the Fourier coefficient. For any vector $\vec{r}$ with orientation $(\theta, \varphi)$, we define $\mathbf{Y}^l(\theta, \psi) = [Y_{-l}^l(\theta, \psi); Y_{-l+1}^l(\theta, \psi); ...; Y_l^l(\theta, \psi)]^T$, a vector with $2l+1$ elements. See details of spherical harmonics in Appendix A.1. Spherical harmonics can be used to encode orientation (Gasteiger et al., 2021; 2022) and map the representation in the frequency domain to the signal over the spatial domain (Cohen et al., 2018; Zitnick et al., 2022).

A commonly used property of the spherical harmonics is that for any $\mathbf{R} \in SO(3)$, we have

$$\mathbf{Y}^l(\mathbf{R}\vec{r}) = \mathbf{D}^l(\mathbf{R})\mathbf{Y}^l(\vec{r}), \tag{3}$$

where $\mathbf{D}^l(\mathbf{R})$ is a $(2l+1) \times (2l+1)$ matrix known as a Wigner-D matrix with the degree $l$. Therefore, $\mathbf{R}$ and $\mathbf{D}^l(\mathbf{R})$ correspond to $T(g)$ and $T'(g)$ in equation 1, respectively. Following the convention in (Chami et al., 2019; Brandstetter et al., 2021), we say that $\mathbf{Y}^l(\vec{r})$ is steerable by the Wigner-D matrix of the same degree $l$. In addition, the $(2l+1)$-dimensional vector space on which a Wigner-D matrix of degree $l$ act is termed a type-$l$ steerable vector space. In fact, the Wigner-D matrix is an irreducible representation of the group $SO(3)$ and we refer interested readers to (Kondor et al., 2018).

## 2.3 EQUIVARIANT OPERATIONS

The key of the equivariant model is to design $\phi(\cdot)$ in equation 1 which preserves the equivariance and at the same time enriches the abstract directional information. A simple approach is to encode the directional information by $\mathbf{Y}^l(\cdot)$ over each node or edge, and then do certain operations preserving equivariance, such as aggregation or scaling with an invariant variable. However, if we depend solely on such operations, it would limit the learning for deep directional features.

The CG tensor product (Griffiths & Schroeter, 2018), originally describing the angular momentum coupling in quantum mechanics, provides another option and becomes the workhorse to design the equivariant networks (Thomas et al., 2018; Kondor et al., 2018; Brandstetter et al., 2021). The CG tensor product $\otimes$ is defined as follows,

$$(\mathbf{u} \otimes \mathbf{v})_m^l = \sum_{m_1=-l_1}^{l_1} \sum_{m_2=-l_2}^{l_2} C_{(l_1,m_1)(l_2,m_2)}^{(l,m)} \mathbf{u}_{m_1}^{l_1} \mathbf{v}_{m_2}^{l_2}, \tag{4}$$

where $\mathbf{u}$ and $\mathbf{v}$ denote two steerable representations with degree $l_1$ and $l_2$. On the right-hand side of equation 4, $C$ denotes the CG coefficients, a sparse tensor, which produces non-zero terms when

$$|l_1 - l_2| \leq l \leq (l_1 + l_2). \tag{5}$$

Based on the CG tensor product, the output is a type-$l$ vector when $\mathbf{u}$ and $\mathbf{v}$ are type-$l_1$ and type-$l_2$ vectors. Besides, there are several equivariant operations in our method, including gate (Weiler et al., 2018; Fuchs et al., 2020) and normalization (Geiger et al., 2022). Their details can be found in Appendix A.2.

## 2.4 ANALYSIS OF EXPRESSIVE POWER

Several studies have indicated that applying the CG tensor product to high-degree representations can push the upper boundary of GNN expressive power. For instance, Theorem 2 in (Dym & Maron, 2021) has demonstrated that any continuous G-equivariant function can be effectively approximated using TFN structures (Thomas et al., 2018) with infinite degrees. Additionally, Joshi et al. (2023) has revealed that equivariant models with a maximum degree $L$ struggle to distinguish $n$-fold symmetric structures when $L$ is less than $n$. In this case, more flexible structures like Multi-Layer Perceptron (MLP) may enhance expressive power, since the functions that MLP approximates encompass universal equivariant functions. We substantiated this perspective through GWL graph isomorphism test (Joshi et al., 2023), as detailed Appendix A.3. Note that these flexible structures introduce non-equivariance. The expected generalization intricately tied to both equivariance and expressiveness. Consequently, our work also focuses on mitigating the loss of equivariance.

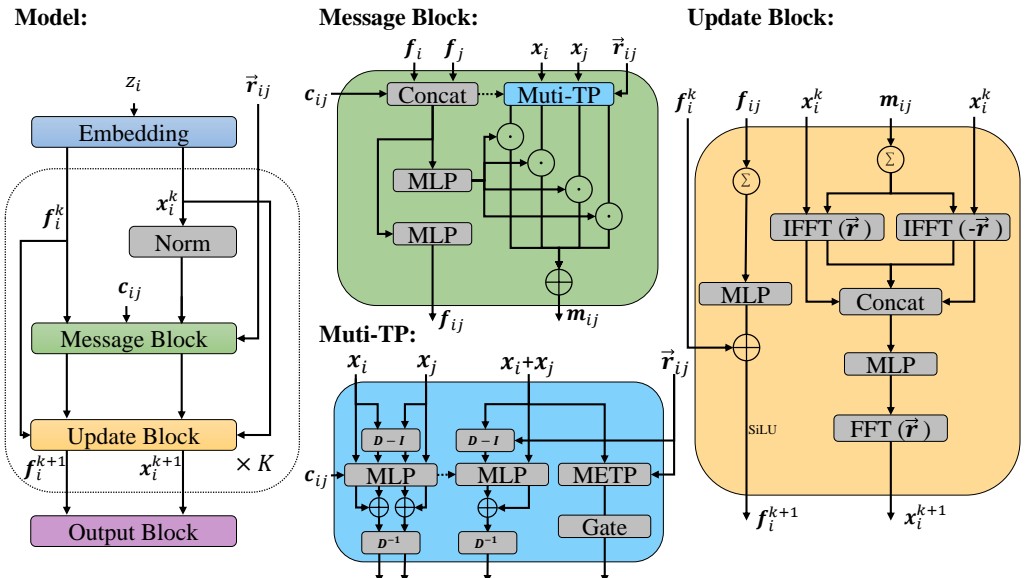

Figure 1: Left: The overall framework of our model; Middle: message block and structure of Muti-TP; Right: update block. Dotted lines with arrow represent the original value of $\mathbf{c}_{ij}$.

## 3 OUR METHOD

The architecture of HDGNN is illustrated in Figure 1. We first introduce the framework of HDGNN. Then we show how to design each ingredient of the neural network.

### 3.1 MESSAGE PASSING FOR MODELING ATOMIC MOLECULAR SYSTEM

Message passing in GNNs is a particularly effective tool to encode the atomic property and model the interactions between the atoms. In our paper, we use the $\vec{\mathbf{r}}_{ij}$ to represent the vector from node $i$ to node $j$. The local neighborhood $\mathcal{N}(i)$ is defined by a cutoff radius $\mathcal{N}(i) = \{j \mid \|\vec{\mathbf{r}}_{ij}\| \le r_{cut}\}$ or the 2D molecular graphs induced by SMILES

HDGNN is built upon the message passing propagation mechanism. To enable steerable node embeddings, we define them as a combination of type-0, type-1, ..., type-$L$ vectors, resulting in a size of $(L+1)^2$. Here, $L$ represents the maximum degree and can be considered a hyper-parameter which we usually set to $[4, 6]$. Similarly, we define a spherical harmonic representation of any 3D vector $\vec{\mathbf{r}}$ as $\mathbf{S}^L(\vec{\mathbf{r}}) = [\mathbf{Y}^0(\vec{\mathbf{r}}); \mathbf{Y}^1(\vec{\mathbf{r}}); \ldots; \mathbf{Y}^L(\vec{\mathbf{r}})]$, which also has a size of $(L+1)^2$. Furthermore, each node embedding is expanded to $C$ channels. In the first round, the invariant atomic feature $z_i$ is passed through an embedding layer to obtain a representation for the $i$-th node. As this representation is invariant, we assign it as the type-0 vector, while the other type-$l$ vectors are set to zero. The overall node embedding is denoted as $\mathbf{x}_i^0$. Then $\mathbf{x}_i^0$ goes through norm layer, message block and update block to incorporate positional information and neighborhood information, which yields a new embedding $\mathbf{x}_i^k$. Additionally, we construct an invariant branch, with its node embedding denoted as $\mathbf{f}_i$. This branch assists in the calculation of messages and the final predictions. The invariant branch, depicted on the left side of the model in Figure 1, is a lightweight structure that ensures invariance of embedding $\mathbf{f}_i$. For further implementation details, please refer to Appendix B.1.

By repeating this procedure $K$ times, we obtain the final embeddings $\mathbf{x}_i^K$ and $\mathbf{f}_i^K$. These embeddings are then processed through specific layers to generate the final results, such as energy, force, or other properties, depending on the specific task at hand. In the following section, we detail each ingredient.

### 3.2 MESSAGE BLOCK

In this section, we will elaborate on the design of the message passing block. Our objective is to compensate for the loss of expressive power resulting from the limitations of finite low-degree terms. To accomplish this, we introduce the message block, depicted in Figure 1, which incorporates both equivariant operations and non-equivariant terms. Formally, the message calculation is defined in

equation 11. To provide insights into the structure of this block, we first present the construction of a strictly equivariant counterpart of equation 11. In this counterpart, we define the message $\mathbf{m}_{ij}$ as follows:

$$\mathbf{m}_{ij} = (\mathbf{x}_i + \mathbf{x}_j) \otimes \mathbf{S}^L(\vec{\mathbf{r}}_{ij}), \tag{6}$$

where $\otimes$ denotes the CG tensor product. In equation 6, we do the product between the node embedding (or messages) and fixed spherical harmonic representations of the edge direction $\vec{\mathbf{r}}_{ij}$, which is the conventional wisdom to implement the CG tensor product in equivariant networks such as (Thomas et al., 2018; Batzner et al., 2022).

We aim to incorporate the higher-degree nonlinearity beyond the bilinear operation of the CG tensor product. To that end, we develop a learnable counterpart of equation 6 to reach this goal. We termed it multiple tensor product (Muti-TP). Each pathway of Muti-TP represents an analogue of CG tensor product, which is shown in Muti-TP module of Figure 1. In the following, we use $\mathbf{D}(\mathbf{R}_{ij}) = \mathbf{D}^0(\mathbf{R}_{ij}) \bigoplus \mathbf{D}^1(\mathbf{R}_{ij}), ..., \bigoplus \mathbf{D}^L(\mathbf{R}_{ij})$ to represent a block diagonal matrix with Wigner-D matrices from degree 0 to $L$. $\mathbf{R}_{ij}$ denotes the 3D rotation matrix that transforms $\vec{\mathbf{r}}_{ij}$ to a fixed orientation $[0, 0, 1]$. redNote that there are multiple rotations $\mathbf{R}_{ij}$ that can rotate $\vec{\mathbf{r}}_{ij}$ to $[0, 0, 1]$. We randomly select one of them and apply it to all message blocks.

To begin with, we introduce the Mean-Extension Tensor Product (METP) as one of the pathways in Multi-TP. The METP is designed to capture equivariant interactions in low-degree representations. For representations in low degrees ($l \leq L'$), where $L'$ is a hyper-parameter typically set to a range of $[1, 4]$. The METP is defined as follows:

$$(\mathbf{u} \otimes_{METP} \mathbf{v})_{mc}^l = w^{lc} \sum_{m_1=-l_1}^{l_1} \sum_{m_2=-l_2}^{l_2} C_{(l_1,m_1)(l_2,m_2)}^{(l,m)} \bar{\mathbf{u}}_{m_1}^{l_1} \bar{\mathbf{v}}_{m_2}^{l_2}, \tag{7}$$

where $w$ denotes the learnable weights. $\bar{\mathbf{u}}$ denotes the mean of the whole channels. It is worth noting that the fully-connected CG tensor product (Fully-TP) (Geiger et al., 2022) is effective but computationally expensive. In contrast, our METP considerably reduces the computational complexity while preserving the property that each output channel depends on all input channels. Furthermore, we apply a gate activation after the METP operation. As previously mentioned, the use of equivariant operations with finite low-degree representations will limit expressiveness. To address this issue, we incorporate a flexible neural structure to mitigate the loss of expressive power. This structure serves as an additional pathway within Multi-TP and is designed to capture approximately equivariant interactions across both low-degree and high-degree representations ($l \leq L$). The structure is illustrated in the Muti-TP module of Figure 1. To understand the principle of this structure, we expand equation 6 to $\mathbf{x}_i \otimes \mathbf{S}^L(\vec{\mathbf{r}}_{ij}) + \mathbf{x}_j \otimes \mathbf{S}^L(\vec{\mathbf{r}}_{ij})$ and each term can be transformed to

$$\mathbf{x}_i \otimes \mathbf{S}^L(\vec{\mathbf{r}}_{ij}) = \mathbf{D}^{-1}(\mathbf{R}_{ij})\big(\mathbf{x}_i' \otimes \mathbf{S}(\vec{\mathbf{C}})\big), \tag{8}$$

where $\mathbf{x}_i' = \mathbf{D}(\mathbf{R}_{ij})\mathbf{x}_i$, $\vec{\mathbf{C}} = [0, 0, 1]$. Hence, the tensor product is simplified to a sparse matrix multiplication, as showcased in (Passaro & Zitnick, 2023). This approach reduces complexity and offers a feasible way for neural networks to learn the equivariance of CG tensor product. Kofinas et al. (2021) and Zitnick et al. (2022) have shown that rotation in local coordinate frames can facilitate the learning of filter in GNNs. To learn the equivariance in equation 8, an intuitive approach is to replace the term $\big(\mathbf{x}_i' \otimes \mathbf{S}(\vec{\mathbf{C}})\big)$ with a MLP, denoted as $NN(\mathbf{x}_i')$. However, equation 8 introduces randomness in rotation $\mathbf{R}_{ij}$, and the general form of MLP is non-equivariant, rendering it unable to compensate for the introduced randomness (see details in Appendix B.2). Therefore, replacing equation 8 with an MLP will break equivariance. In order to mitigate the loss of equivariance, we propose an alternative approach, which first transform the equation 8 to

$$\mathbf{x}_i \otimes \mathbf{S}^L(\vec{\mathbf{r}}_{ij}) = \mathbf{D}^{-1}(\mathbf{R}_{ij})\big(\mathbf{x}_i'' \otimes \mathbf{S}(\vec{\mathbf{C}}) + \mathbf{x}_i \otimes \mathbf{S}(\vec{\mathbf{C}})\big), \tag{9}$$

where $\mathbf{x}_i'' = (\mathbf{D}(\mathbf{R}_{ij}) - \mathbf{I})\mathbf{x}_i$, and $\mathbf{I}$ denotes an identity matrix. Then, we use a MLP $NN(\mathbf{x})$ to approximate the equivariant function $\big(\mathbf{x} \otimes \mathbf{S}(\vec{\mathbf{C}}_{ij})\big)$. Thus, equation 9 is approximated by

$$\mathbf{x}_i \otimes \mathbf{S}^L(\vec{\mathbf{r}}_{ij}) \approx \mathbf{D}^{-1}(\mathbf{R}_{ij})\big(NN(\mathbf{x}_i'') + NN(\mathbf{x}_i)\big). \tag{10}$$

The weights of $NN(\mathbf{x}_i'')$ and $NN(\mathbf{x}_i)$ are shared. If the neural network $NN(\cdot)$ can learn the complete equivariance by training, the right side of equation 10 is equivariant. We adopt this structure for two objectives: (a) The neural network learns the SO(3)-transformation pattern during training to acquire learned equivariance. We introduce an additional pattern in equation 9 to enhance the quality

of learned equivariance. (b) By utilizing $\mathbf{x}_i$ and $\mathbf{x}_i''$ as independent inputs for MLP, we anticipate that MLP will effectively extract directional information embedded in the Wigner-D matrix.

Based on structures in equation 7 and equation 10, we incorporate an attention module to regulate the contribution of different terms. Combining all pieces together, we have

$$
\begin{aligned}
\mathbf{m}_{ij} = {} & \mathbf{a}_{ij,1} \odot \mathbf{D}^{-1}(\mathbf{R}_{ij})\big(NN_1(\mathbf{x}_i'') + NN_1(\mathbf{x}_i)\big) + \mathbf{a}_{ij,2} \odot \mathbf{D}^{-1}(\mathbf{R}_{ij})\big(NN_1(\mathbf{x}_j'') + NN_1(\mathbf{x}_j)\big) \\
& + \mathbf{a}_{ij,3} \odot \mathbf{D}^{-1}(\mathbf{R}_{ij})\big(NN_2(\mathbf{x}_i'' + \mathbf{x}_j'') + NN_2(\mathbf{x}_i + \mathbf{x}_j)\big) \\
& + \mathbf{a}_{ij,4} \odot (\mathbf{x}_i + \mathbf{x}_j) \otimes_{METP} \mathbf{S}^L(\vec{\mathbf{r}}_{ij}),
\end{aligned}
\tag{11}
$$

where all the attention coefficients $\mathbf{a}_{ij}$ are produced by applying a MLP on invariant branch. The addition in equation equation 11 corresponds to an degree-wise addition. In equation 11, the first three terms, designed to enhance expressive power, are initially non-equivariant. However, through effective training, they tend to achieve approximate equivariance. The last term is inherently equivariant, ensuring that global equivariance is not excessively compromised during training.

In our implementation, we utilize a 2-layer MLP with the SiLU activation function. Additionally, we incorporate two strategies into the MLP structure. First, we introduce built-in molecular properties by multiplying invariant features with the output of the first MLP layer. These features, represented as $\mathbf{c}_{ij}$ in Figure 1, include atomic type embeddings ($z_i$ and $z_j$) and edge attributes obtained from a distance block using a Gaussian basis. For more detailed discussions on the implementation, please refer to Appendix B.2.

We conduct an ablation study on the structures of non-equivariant terms in equation 11, in which we compare two extreme cases with our design. In the first case, we replace the MLP with the (equivariant) linear layer and therefore equation 10 reduces to an equivariant term. In the second case, we naively create a high-degree message $\mathbf{m}_{ij}$ by $NN([\mathbf{x}_i; \mathbf{x}_j; \mathbf{S}(\vec{\mathbf{r}}_{ij})])$, which violates the equivariance a lot and hurts the performance.

### 3.3 UPDATE BLOCK

After aggregating the messages $\mathbf{m}_{ij}$ for each node to gain $\mathbf{m}_i$, our focus shifts to learning the interactions between the channels within each $\mathbf{m}_i$. Using CG tensor products between different channels is a naive approach to ensure equivariance. However, this approach is computationally expensive and falls short in effectively capturing interactions across multiple channels ($\gg 2$). Here, we apply a more efficient method inspired by the observation that certain paths of the CG tensor product resemble convolution or correlation operations in the frequency domain, with the degree $l$ serving as an analog to the frequency $\omega$. To formalize this, let $\mathbf{m}_1$ and $\mathbf{m}_2$ represent any two channels within $\mathbf{m}$. The correlation $\star$ are represented by $(\mathbf{m}_1 \star \mathbf{m}_2)^l = \sum_{dl} \mathbf{m}_1^{dl} \mathbf{m}_2^{dl-l}$, where $(dl, dl-l, l)$ corresponds to no-zero paths $|l_1 - l_2| = l$ in equation 5. Similarly, convolution corresponds to paths $l_1 + l_2 = l$. Inspired by the convolution theorem of the Fourier Transform (FT), we can approximate the tensor product in the frequency domain by performing point-wise operations in the time domain. To this end, we apply the spherical harmonics expansion, which is analogous to the inverse fast Fourier Transform (IFFT), to each channel of $\mathbf{m}$:

$$
p_c(\vec{\mathbf{r}}) = \sum_{l=0}^{L} \sum_{m=-l}^{l} \mathbf{m}_{cm}^l Y_m^l(\vec{\mathbf{r}}).
\tag{12}
$$

Next, we concatenate all the elements in the points $\vec{\mathbf{r}}$ and $-\vec{\mathbf{r}}$ and obtain $\mathbf{P}(\vec{\mathbf{r}}) = [p_1(\vec{\mathbf{r}}), \ldots, p_C(\vec{\mathbf{r}}), p_1(-\vec{\mathbf{r}}), \ldots, p_C(-\vec{\mathbf{r}})]$. Here, we extend the point-wise operation to two points by incorporating both convolution and correlation. Then we apply a MLP (shown in Appendix B.3) with an output size of $C$ to $\mathbf{P}(\vec{\mathbf{r}})$ and gain $\mathbf{P}'(\vec{\mathbf{r}})$. Finally, we transform the results back to the frequency domain:

$$
\mathbf{m}_{cm}^l = \int_{\Omega} \mathbf{P}'(\vec{\mathbf{r}})_c Y_{m*}^l(\vec{\mathbf{r}}) d\vec{\mathbf{r}},
\tag{13}
$$

where $Y_{m*}^l$ denotes the complex conjugation. In our implementation, we use the Fast Fourier Transform (FFT) on $S^2$ to represent the integral. Additionally, we incorporate a shortcut where we transform the input of the message block, $\mathbf{x}^k$, to the time domain and concatenate the results with $\mathbf{P}(\vec{\mathbf{r}})$ before passing them through the MLP. In this case, the output of equation 13 is the new embedding $\mathbf{x}^{k+1}$. Our approach shares similarities with (Cohen et al., 2018; Zitnick et al., 2022), where complex operations are transformed to other domains based on the convolution theorem. They apply a single transformation, whereas we combine both correlation and convolution. Our approach

enhances the paths of the CG tensor product embedded in time domain signals, thereby facilitating effective learning of interactions between channels. Our ablation experiments provide support for the efficacy of our approach. These experiments and theoretical details can be found Appendix B.3.

## 3.4 OUTPUT BLOCK

The final message passing block generates per-atom features $\mathbf{x}^K$. To predict chemical properties, we adopt the approach outlined in (Zitnick et al., 2022). Optionally, we include the output of the invariant branch, $\mathbf{f}^K$, in the prediction process. For some invariant properties, we add $\lambda \cdot NN(\mathbf{f}^K)$ to the prediction produced by $\mathbf{x}^K$. Here, $\lambda$ is a fixed weight.

## 4 RELATED WORKS

We focus on equivariant models closely related to ours here. We provide a detailed discussion of other molecular models and works on relaxing equivariance in Appendix C.

Recent research showed that equivariant message passing neural network have achieved remarkable results in predicting invariant or equivariant molecular properties (Thomas et al., 2018; Fuchs et al., 2020; Batzner et al., 2022; Brandstetter et al., 2021). TFN (Thomas et al., 2018) and NequIP (Batzner et al., 2022) define the convolution filters $F^{l,l'}(\vec{\mathbf{r}}_{ij}) = R^{l,l'}(\|\vec{\mathbf{r}}_{ij}\|)\mathbf{Y}^l(\frac{\vec{\mathbf{r}}_{ij}}{\|\vec{\mathbf{r}}_{ij}\|})$, where $R^{l,l'}(\cdot)$ is a learnable radial function. They define the messages as $\mathbf{m}_{ij} = F^{l,l'}(\vec{\mathbf{r}}_{ij}) \otimes \mathbf{x}_i^{l'}$, where $\mathbf{x}_i^{l'}$ is a type $l'$ steerable vector of node $i$. They provide a general framework to combine spherical harmonics basis and embedding. Utilizing a spherical harmonics basis enables the learning of intricate equivariant functions (Weiler et al., 2018; Dym & Maron, 2021). However, the expressiveness of these methods is constrained by a band-limited degree (Cesa et al., 2021). SE(3)-transformer (Fuchs et al., 2020) introduces an attention weight $a_{ij}$ to enhance the expressiveness of the network. Specifically, the message can be abstracted to $\mathbf{m}_{ij} = a_{ij}F^{l,l'}(\vec{\mathbf{r}}_{ij}) \otimes \mathbf{x}_i^{l'}$. Yet, the attention weight is a scalar or an invariant term independent of the CG tensor product. SEGNN (Brandstetter et al., 2021) and Equiformer (Liao & Smidt, 2023) extend the function form of above-mentioned works by incorporating the message passing neural networks and transformer-based networks, while its workhorse is still convolution filter with the CG tensor product, causing the similar limitation of TFN and SE(3)-transformer. In our work, we do not overconstrain the function form of the convolution filter. Notice that Wigner-D matrices in the message block contain the information of spherical harmonics. We hope that the network can learn interactions between $\mathbf{x}_i$ and $Y_m^l(\vec{\mathbf{r}}_i)$, which may replace the CG tensor product. SCN (Zitnick et al., 2022) employs non-equivariant operations in molecular models. Despite sharing the concept of relaxing equivariance, our structures that used to calculate messages are entirely different. Moreover, the equivariance of SCN heavily relies on data, which may be problematic on small datasets. In contrast, our method addresses the loss of equivariance through strict equivariant operations.

## 5 EXPERIMENTS

In this section, we conduct experiments to investigate the effectiveness of proposed method over Quantum Machines 9 (QM9) (Ramakrishnan et al., 2014) and IS2RE task in Open Catalyst 2020 (OC20) (Chanussot et al., 2021) benchmarks. In both experiments, we include Equiformer (Liao & Smidt, 2023), SEGNN (Brandstetter et al., 2021) and TFN (Thomas et al., 2018), strong baselines of equivariant neural network; Dimenet++ (Klicpera et al., 2020), strong baselines of directional GNNs; Schnet (Schütt et al., 2018) and PaiNN (Schütt et al., 2021), classical networks for modeling quantum interactions. In the task of IS2RE, we include an additional baseline GemNet (Gasteiger et al., 2021; 2022), SphereNet (Liu et al., 2021) and SCN (Zitnick et al., 2022). In the task of QM9, L1Net (Miller et al., 2020), Cormorant (Anderson et al., 2019), LieConv (Finzi et al., 2020), TorchMD-NET (Thölke & Fabritiis, 2022) and EQGAT (Le et al., 2022) are also compared with our work. We defer details of configurations and hyper-parameters of baselines to Appendix D.

## 5.1 RESULTS

The OC20 dataset contains over 130 million structures used to train models for predicting forces and energies during structure relaxations with a CC Attribution 4.0 License. The goal of a structure relaxation is to find a local energy minimum. We report results of Initial Structure to Relaxed Energy task (IS2RE). The IS2RE can be solved by two approaches: 1.directly predict relaxed energy from initial atomic structure; 2.use the relaxed structure computed by predicted forces to predict energy. The second approach tends to be more accurate, while it needs an efficient force model trained from other bigger dataset. We focus on the first approach without pre-trained models as that in (Brandstetter

Table 1: Results on IS2RE OC20 Test for approaches that directly predict the relaxed energies.

| Model | Energy MAE (meV) ↓ | | | | | EwT (%) ↑ | | | |
|---|---|---|---|---|---|---|---|---|---|
| | ID | OOD Ads | OOD Cat | OOD Both | Average | ID | OOD Ads | OOD Cat | OOD Both |
| Median baseline | 1750 | 1879 | 1709 | 1664 | 1750 | 0.71 | 0.72 | 0.89 | 0.74 |
| CGCNN | 615 | 916 | 622 | 851 | 751 | 3.40 | 1.93 | 3.10 | 2.00 |
| SchNet | 639 | 734 | 662 | 704 | 685 | 2.96 | 2.33 | 2.94 | 2.21 |
| PaiNN | 575 | 783 | 604 | 743 | 676 | 3.46 | 1.97 | 3.46 | 2.28 |
| TFN (SElin) | 584 | 766 | 636 | 700 | 672 | 4.32 | 2.51 | 4.55 | 2.66 |
| GemNet-dT | 527 | 758 | 549 | 702 | 634 | 4.59 | 2.09 | 4.47 | 2.28 |
| DimeNet++ | 562 | 725 | 576 | 661 | 631 | 4.25 | 2.07 | 4.10 | 2.41 |
| GemNet-OC | 560 | 711 | 576 | 671 | 630 | 4.15 | 2.29 | 3.85 | 2.28 |
| SphereNet | 563 | 703 | 571 | 638 | 619 | 4.47 | 2.29 | 4.09 | 2.41 |
| SEGNN | 533 | 692 | 537 | 679 | 610 | **5.37** | 2.46 | **4.91** | 2.63 |
| Equiformer | **504** | 688 | **521** | 630 | 586 | 5.14 | 2.41 | 4.67 | 2.69 |
| SCN | 516 | 643 | 530 | 604 | 573 | 4.92 | 2.71 | 4.42 | 2.76 |
| HDGNN | 510 | **618** | 523 | **550** | **548** | 5.08 | **2.79** | 4.58 | **2.82** |

et al., 2021). The test set is splited into four subsets, including sampling from the same distribution as training (ID), unseen adsorbates (OOD Ads), unseen element compositions for catalysts (OOD Cat), and unseen adsorbates and catalysts (OOD Both). Their sizes are similar. As shown in Table 1, HDGNN outperforms all previous approaches in terms of the average energy MAE. Furthermore, a notable observation is that HDGNN obviously outperforms other methods on unseen distribution, as evidenced by the significantly lower average MAE on OOD tasks (HDGNN: 564 vs. SCN/Equiformer: 592/613). We hypothesize that the enhanced performance of HDGNN on unseen distributions may stem from learned features that closely align with actual physical properties. Additionally, our model achieves the best results in terms of the EwT metric (percentage of predictions within $\epsilon = 0.02$ eV of the ground truth) among OOD Ads and OOD Both.

Table 2: Results on QM9 dataset for various chemical properties. † denotes using different data partitions. Bold and underline indicate the best result, and the second best result, respectively.

| Task | $\alpha$ | $\Delta$ | $\varepsilon_{HOMO}$ | $\varepsilon_{LUMO}$ | $\mu$ | $C_v$ | $G$ | $H$ | $R^2$ | $U$ | $U_0$ | ZPVE |
| Units | bohr$^3$ | meV | meV | meV | D | cal/(mol K) | meV | meV | bohr$^3$ | meV | meV | meV |
|---|---|---|---|---|---|---|---|---|---|---|---|---|
| SchNet | .235 | 63 | 41 | 34 | .033 | .033 | 14 | 14 | .073 | 19 | 14 | 1.70 |
| Cormorant† | .085 | 61 | 34 | 38 | .038 | .026 | 20 | 21 | .961 | 21 | 22 | 2.02 |
| L1Net | .088 | 68 | 46 | 35 | .043 | .031 | 14 | 14 | .354 | 14 | 13 | 1.56 |
| LieConv† | .084 | 49 | 30 | 25 | .032 | .038 | 22 | 24 | .800 | 19 | 19 | 2.28 |
| TFN† | .223 | 58 | 40 | 38 | .064 | .101 | - | - | - | - | - | - |
| DimeNet++ | **.044** | 33 | 25 | 20 | .030 | **.023** | 8 | 7 | .331 | 6 | 6 | **1.21** |
| PaiNN | .045 | 46 | 28 | 20 | .012 | .024 | **7.35** | **5.98** | .066 | **5.83** | **5.85** | 1.28 |
| TorchMD-NET | .059 | 36 | 20 | 18 | **.011** | .026 | 7.62 | 6.16 | **.033** | 6.38 | 6.15 | 1.84 |
| SEGNN† | .060 | 42 | 24 | 21 | .023 | .031 | 15 | 16 | .660 | 13 | 15 | 1.62 |
| EQGAT | .053 | 32 | 20 | 16 | **.011** | .024 | 23 | 24 | .382 | 25 | 25 | 2.00 |
| Equiformer | .046 | **30** | **15** | **14** | **.011** | **.023** | 7.63 | 6.63 | .251 | 6.74 | 6.59 | 1.26 |
| HDGNN | .046 | 32 | 18 | 16 | .017 | **.023** | 11 | 10 | .342 | 8.12 | 8.34 | **1.21** |

The QM9 benchmark provides quantum chemical properties for a relevant, consistent, and comprehensive chemical space of $134k$ stable small organic molecules up to 29 atoms. Each atom is described with 3D position coordinates and embedding of its atomic type (H, C, N, O, F). The QM9 benchmark is a regression task where we optimize the MAE between multiple chemical properties and their ground truths. Here, we replace METP with a Fully-TP because the low-order Fully-TP on small molecular system can not cause much computational burden. From Table 2, our model achieves the best results on two tasks and the second best results on three tasks. Note that we employ the same architecture for all tasks, while TorchMD-NET and PaiNN utilize different architectures for the $\mu$ and $R^2$ tasks, potentially leading to a more accurate inductive basis. Additionally, HDGNN falls behind the SOTA methods on the energy variables ($G, H, U, U_0$) tasks. This may be attributed to the fact that such targets may benefit from more architectures including attention mechanisms, neighbor-neighbor interactions and problem-tailored architectures (Brandstetter et al., 2022) used in the comparative approaches.

The QM9 dataset is considerably smaller than OC20, rendering models susceptible to overfitting during learning equivariance. For instance, SCN struggles to achieve SOTA results on QM9 tasks due to its reliance on learning equivariance, unlike other QM9 leading methods that possess inherent equivariance. In contrast, our approach merges inherent equivariance with learned equivariance, empowering it to excel in QM9 tasks and achieve SOTA results.

## 5.2 ABLATION STUDY

In this section, we explore several pivotal questions: 1. Can approximatively equivariant operations achieve improved performance compared to equivariant operations? 2. Does the attention module contribute to regulating both expressiveness and equivariance? 3. Are sub-structures or design in HDGNN valid? The rest ablation experiments can be found in Appendix E, which investigate the following aspects: 1) the effectiveness of normalization and the invariant branch, 2) the structure of each MLP in HDGNN, 3) the effectiveness of each component in the update block, and 4) the impact of hyperparameters. Furthermore, we analyze the training and inference times.

To begin with, we compare the strict equivariance and approximate equivariance. While HDGNN has demonstrated competitive results compared to strictly equivariant models in certain molecular tasks, it is important to determine whether these improvements are attributed to the non-equivariant design. To address this, we construct two analogue equivariant models of HDGNN ("Linear" and "Fully-TP" in Table 3). One model replaces the neural networks with equivariant linear layers, while the other only uses fully-connected tensor product operations instead of equation 11. Besides, we construct a model that uses $NN([\mathbf{x}_i; \mathbf{x}_j; \mathbf{S}(\vec{\mathbf{r}}_{ij})])$ to calculate messages ("Unrestricted"). The baseline denotes a HDGNN where $K = 8$, $C = 64$ and $L$=6. We have three observations from Table 3: 1.although the linear structure is equivariant, it fails to capture the fine-grained interaction between atoms; 2.approximately equivariant model can achieve the better generalization compared to equivariant model; 3.Designing approximately equivariant models requires careful consideration, as unconstrained learning modules can significantly undermine the generalization abilities of the models.

Table 3: Ablation studies for equivariance.

| Model | Energy MAE (meV) ↓ | | | | |
| --- | --- | --- | --- | --- | --- |
| | ID | OOD Ads | OOD Cat | OOD Both | Average |
| Baseline | **554** | **701** | **566** | **642** | **616** |
| (Linear) | 632 | 748 | 664 | 711 | 689 |
| (Fully-TP) | 591 | 752 | 646 | 704 | 673 |
| (Unrestricted) | 883 | 1052 | 962 | 996 | 973 |

Table 4: Ablation studies for attention module.

| Model | Train MAE ↓ | MAD ↓ | Test MAE |
| --- | --- | --- | --- |
| (Sigmoid) | 513 | 19.9 | 554 |
| (SoftMax) | 520 | 20.8 | 562 |
| (0.25, 0.25, 0.25 0.25) | 510 | 21.5 | 564 |
| (0.3, 0.3, 0.3, 0.1) | 511 | 21.9 | 567 |

The attention module in HDGNN is crucial since it can adaptively adjust the equivariance of model. To evaluate it, we use mean absolute difference (MAD), training MAE and test MAE to approximately denote equivariance, expressiveness and generalization, respectively. Note that MAD we used is equal to $|E - E'|$, where $E$ is the predicted energy and $E'$ is the predicted energy based on random SO(3) transformation. We remove the attention module of message block or replace the output activation of attention module to SoftMax. As shown in Table 4, the attention module with Sigmoid achieves the best result. Note that the last two rows in Table 4 represent using fixed weights to replace $\mathbf{a}_1$, $\mathbf{a}_2$, $\mathbf{a}_3$, $\mathbf{a}_4$ in equation 11. Additionally, we manually adjust the attention coefficients to find the trade-off between equivariance and expressiveness on QM9 task. The detail can be found in Appendix E.5.

Table 5: Ablation studies for non-equivariant module in HDGNN.

| Model | Layer Norm | No Linear | Shared MLP | $NN(\mathbf{x}'')$ | MAE (meV) ↓ |
| --- | --- | --- | --- | --- | --- |
| | ✓ | ✓ | ✓ | ✓ | 554 |
| | - | ✓ | ✓ | ✓ | 577 |
| HDGNN | ✓ | - | ✓ | ✓ | 568 |
| | ✓ | ✓ | - | ✓ | 555 |
| | ✓ | ✓ | ✓ | - | 572 |

At last, we investigate our approximate equivariant module by several experiments: 1.removing layer norm for $\mathbf{x}_i$; 2.introducing (equivariant) linear layer for $\mathbf{x}_i$ and $\mathbf{x}_j$ in message block; 3.using different neural networks in equation 11; 4.transforming $NN(\mathbf{x}'') + NN(\mathbf{x})$ to $NN(\mathbf{x}')$ in equation 11. We have two main observations from Table 5, one of which is that redundant linear layers do not lead to improved performance. More importantly, replacing our operation in learnable module will cause a performance degradation.

## 6 CONCLUSION AND FUTURE WORK

In this paper, we offer a new possible solution to predict the properties of molecules beyond the strictly equivariant neural networks and demonstrate its superiority over the QM9 benchmark and IS2RE dataset of OC20. One limitation is related to the hyper-parameter $L'$. Increasing $L'$ to the higher order may bring extra benefit but leads to more computations. Hence, one direction of the future work is to systematically analyze the effect of $L'$.

## ACKNOWLEDGEMENTS

We are thankful to the anonymous reviewers for their helpful comments. This work was supported in part by the STI 2030-Major Projects of China under Grant 2021ZD0201300, and by the National Science Foundation of China under Grant 62276127. The corresponding authors are Chao Qu and Furao Shen.

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
