APPENDIX

## A  THE MATHEMATICS

### A.1  THE MATHEMATICS OF SPHERICAL HARMONICS

#### A.1.1  THE PROPERTIES OF SPHERICAL HARMONICS

The spherical harmonics $Y_l^m(\theta, \phi)$ are the angular portion of the solution to Laplace's equation in spherical coordinates where azimuthal symmetry is not present. Some care must be taken in identifying the notational convention being used. In this entry, $\theta$ is taken as the polar (colatitudinal) coordinate with $\theta$ in $[0, \pi]$, and $\phi$ as the azimuthal (longitudinal) coordinate with $\phi$ in $[0, 2\pi)$.

Spherical harmonics satisfy the spherical harmonic differential equation, which is given by the angular part of Laplace's equation in spherical coordinates. If we define the solution of Laplace's equation as $F = \Phi(\phi)\Theta(\theta)$, the equation can be transformed as:

$$\frac{\Phi(\phi)}{\sin\theta}\frac{d}{d\theta}\left(\sin\theta\frac{d\Theta}{d\theta}\right) + \frac{\Theta(\theta)}{\sin^2\theta}\frac{d^2\Phi(\phi)}{d\phi^2} + l(l+1)\Theta(\theta)\Phi(\phi) = 0 \tag{14}$$

Here we omit the derivation process and just show the result. The (complex-value) spherical harmonics are defined by:

$$Y_m^l(\theta, \phi) \equiv \sqrt{\frac{2l+1}{4\pi}\frac{(l-m)!}{(l+m)!}}P_m^l(\cos\theta)e^{im\phi}, \tag{15}$$

where $P_m^l(\cos\theta)$ is an associated Legendre polynomial. Spherical harmonics are integral basis, which satisfy:

$$
\begin{aligned}
&\int_0^{2\pi}\int_0^\pi Y_{m_1}^{l_1}(\theta,\phi)Y_{m_2}^{l_2}(\theta,\phi)Y_{m_3}^{l_3}(\theta,\phi)\sin\theta d\theta d\phi \\
&= \sqrt{\frac{(2l_1+1)(2l_2+1)(2l_3+1)}{4\pi}}\begin{pmatrix} l_1 & l_2 & l_3 \\ 0 & 0 & 0 \end{pmatrix}\begin{pmatrix} l_1 & l_2 & l_3 \\ m_1 & m_2 & m_3 \end{pmatrix},
\end{aligned}
\tag{16}
$$

where $\begin{pmatrix} l_1 & l_2 & l_3 \\ m_1 & m_2 & m_3 \end{pmatrix}$ is a Wigner 3j-symbol (which is related to the Clebsch-Gordan coefficients). We list a few spherical harmonics which are:

$$
\begin{aligned}
Y_0^0(\theta,\varphi) &= \frac{1}{2}\sqrt{\frac{1}{\pi}} \\
Y_{-1}^1(\theta,\varphi) &= \frac{1}{2}\sqrt{\frac{3}{2\pi}}\sin\theta e^{-i\varphi} \\
Y_0^1(\theta,\varphi) &= \frac{1}{2}\sqrt{\frac{3}{\pi}}\cos\theta \\
Y_1^1(\theta,\varphi) &= \frac{-1}{2}\sqrt{\frac{3}{2\pi}}\sin\theta e^{i\varphi} \\
Y_{-2}^2(\theta,\varphi) &= \frac{1}{4}\sqrt{\frac{15}{2\pi}}\sin^2\theta e^{-2i\varphi} \\
Y_{-1}^2(\theta,\varphi) &= \frac{1}{2}\sqrt{\frac{15}{2\pi}}\sin\theta\cos\theta e^{-i\varphi} \\
Y_0^2(\theta,\varphi) &= \frac{1}{4}\sqrt{\frac{5}{\pi}}\left(3\cos^2\theta - 1\right) \\
Y_1^2(\theta,\varphi) &= \frac{-1}{2}\sqrt{\frac{15}{2\pi}}\sin\theta\cos\theta e^{i\varphi} \\
Y_2^2(\theta,\varphi) &= \frac{1}{4}\sqrt{\frac{15}{2\pi}}\sin^2\theta e^{2i\varphi}
\end{aligned}
\tag{17}
$$

In this work, we use the real-value spherical harmonics rather than the complex-value one.

### A.1.2 FOURIER TRANSFORMATION OVER $S^2$

It is well known that the spherical harmonic $Y_m^l$ form a complete set of orthonormal functions and thus form an orthonormal basis of the Hilbert space of square-integrable function. On the unit sphere $S^2$, any square-integrable function $f$ can thus be expanded as a linear combination of these:

$$
f(\theta,\varphi) = \sum_{l=0}^\infty\sum_{m=-l}^l f_m^l Y_m^l(\theta,\varphi),
\tag{18}
$$

The coefficient $f_m^l$ can be obtained by the Fourier transformation over $S^2$, which is

$$
f_m^l = \int_{S^2} f(\vec{r})Y_m^{l*}(\vec{r})d\vec{r} = \int_0^{2\pi}\int_0^\pi d\theta\sin\theta f(\theta,\psi)Y_m^{l*}(\theta,\psi).
\tag{19}
$$

Usually we define a vector $\mathbf{f}^l = [f_{-l}^l, f_{-l+1}^l, ..., f_l^l]$ to denote the Fourier coefficients with degree $l$. We now investigate how the fourier coefficients transforms if we rotate the input signal. More precisely, we want to calculate the coefficient $\mathbf{f}_\mathbf{R}^l$ of the signal $f(\mathbf{R}\vec{r})$, where $\mathbf{R}\in SO(3)$ is a rotation matrix.

Using the fact $\mathbf{Y}^l(\mathbf{R}\vec{r}) = \mathbf{D}^l(\mathbf{R})\mathbf{Y}^l(\vec{r})$, and equation 20, we know

$$
f(\mathbf{R}\vec{r}) = \sum_{l=0}^\infty\sum_{m=-l}^l f_m^l Y_m^l(\mathbf{R}\vec{r}) = \sum_{l=0}^\infty\sum_{m=-l}^l f_m^l\sum_{m'}\mathbf{D}_{mm'}Y_{m'}^l(\vec{r}).
$$

Therefore $\mathbf{f}_{\mathbf{R}}^l = \mathbf{D}^T \mathbf{f}^l$ and it is steerable.

It is well known that the spherical harmonic $Y_m^l$ form a complete set of orthonormal functions and thus form an orthonormal basis of the Hilbert space of square-integrable function. On the unit sphere $S^2$, any square-integrable function $f$ can thus be expanded as a linear combination of these:

$$f(\theta, \varphi) = \sum_{l=0}^{\infty} \sum_{m=-l}^{l} f_m^l Y_m^l(\theta, \varphi), \tag{20}$$

The coefficient $f_m^l$ can be obtained by the Fourier transformation over $S^2$, which is

$$f_m^l = \int_{S^2} f(\vec{r}) Y_m^{l*}(\vec{r}) d\vec{r} = \int_0^{2\pi} \int_0^{\pi} d\theta \sin \theta f(\theta, \psi) Y_m^{l*}(\theta, \psi). \tag{21}$$

Usually we define a vector $\mathbf{f}^l = [f_{-l}^l, f_{-l+1}^l, ..., f_l^l]$ to denote the Fourier coefficients with degree $l$. We now investigate how the fourier coefficients transforms if we rotate the input signal. More precisely, we want to calculate the coefficient $\mathbf{f}_{\mathbf{R}}^l$ of the signal $f(\mathbf{R}\vec{r})$, where $\mathbf{R} \in SO(3)$ is a rotation matrix.

Using the fact $\mathbf{Y}^l(\mathbf{R}\vec{r}) = \mathbf{D}^l(\mathbf{R})\mathbf{Y}^l(\vec{r})$, and equation 20, we know

$$f(\mathbf{R}\vec{r}) = \sum_{l=0}^{\infty} \sum_{m=-l}^{l} f_m^l Y_m^l(\mathbf{R}\vec{r}) = \sum_{l=0}^{\infty} \sum_{m=-l}^{l} f_m^l \sum_{m'} \mathbf{D}_{mm'} Y_{m'}^l(\vec{r}).$$

Therefore $\mathbf{f}_{\mathbf{R}}^l = \mathbf{D}^T \mathbf{f}^l$ and it is steerable.

### A.1.3 THE RELATIONSHIP BETWEEN SPHERICAL HARMONICS AND WIGNER-D MATRIX

A rotation $\mathbf{R}$ sending the $\vec{r}$ to $\vec{r}'$ can be regarded as a linear combination of spherical harmonics that are set to the same degree. The coefficients of linear combination represent the complex conjugate of an element of the Wigner D-matrix. The rotational behavior of the spherical harmonics is perhaps their quintessential feature from the viewpoint of group theory. The spherical harmonics $Y_m^l$ provide a basis set of functions for the irreducible representation of the group SO(3) with dimension $(2l+1)$.

The Wigner-D matrix can be constructed by spherical harmonics. Consider a transformation $Y_m^l(\vec{r}) = Y_m^l(\mathbf{R}_{\alpha, \beta, \gamma} \vec{r}_x)$, where $\vec{r}_x$ denote the x-orientation. $\alpha, \beta, \gamma$ denotes the items of Euler angle. Therefore, $Y_m^l(\vec{r})$ is invariant with respect to rotation angle $\gamma$. Based on this viewpoint, the Wigner-D matrix with shape $(2l+1) \times (2l+1)$ can be defined by:

$$D_m^l(\mathbf{R}_{\alpha, \beta, \gamma}) = \sqrt{2l+1} Y_m^l(\vec{r}). \tag{22}$$

In this case, the orientations are encoded in spherical harmonics and their Wigner-D matrices, which are utilized in our cross module.

## A.2 EQUIVARIANT OPERATION

### A.2.1 EQUIVARIANCE OF CLEBSCH-GORDAN TENSOR PRODUCT

The Clebsch-Gordan Tensor Product shows a strict equivariance for different group representations, which make the mixture representations transformed equivariant based on Wigner-D matrices. We use $D_{m'_1, m_1}$ to denote the element of Wigner-D matrix. The Clebsch-Gordan coefficient satisfies:

$$\begin{aligned} \sum_{m'_1, m'_2} C_{(l_1, m'_1)(l_2, m'_2)}^{(l_0, m_0)} D_{m'_1 m_1}^{l_1}(g) D_{m'_2 m_2}^{l_2}(g) \\ = \sum_{m'_0} D_{m_0 m'_0}^{l_0}(g) C_{(l_1, m_1)(l_2, m_2)}^{(l_0, m'_0)} \end{aligned} \tag{23}$$

Therefore, the spherical harmonics can be combined equivariantly by CG Tensor Product:

$$
\begin{aligned}
& CG\left(\sum_{m'_1} D^{l_1}_{m_1 m'_1}(g) Y^{l_1}_{m'_1}, \sum_{m'_2} D^{l_2}_{m_2 m'_2}(g) Y^{l_2}_{m'_2}\right)^{l_0}_{m_0} \\
&= \sum_{m_1,m_2} C^{(l_0,m_0)}_{(l_1,m_1)(l_2,m_2)} \sum_{m'_1} D^{l_1}_{m_1 m'_1}(g) Y^{l_1}_{m'_1} \sum_{m'_2} D^{l_2}_{m_2 m'_2}(g) Y^{l_2}_{m'_2} \\
&= \sum_{m'_0} D^{l_o}_{m_0 m'_0}(g) \sum_{m_1,m_2} C^{(l_0,m'_0)}_{(l_1,m_1)(l_2,m_2)} Y^{l_1}_{m'_1} Y^{l_2}_{m'_2} \\
&= \sum_{m'_0} D^{l_0}_{m_0 m'_0}(g) CG^{l_0}_{m'_0}\left(Y^{l_1}_{m'_1}, Y^{l_2}_{m'_2},\right).
\end{aligned}
\tag{24}
$$

equation 24 represents a relationship between scalar. If we transform the scalar to vector or matrix like what we do in Section 2 and Section 3, equation 24 is equal to

$$
(\mathbf{D}^{l_1}_{\mathbf{R}} \mathbf{u} \otimes \mathbf{D}^{l_2}_{\mathbf{R}} \mathbf{v})^l = \mathbf{D}^l_{\mathbf{R}}(\mathbf{u} \otimes \mathbf{v})^l.
\tag{25}
$$

The tensor CG product mixes two representations to a new representation under special rule equation 5. For example, 1.two type-0 vectors will only generate a type-0 representations; 2.type-$l_1$ and type-$l_2$ can generate type-$l_1 + l_2$ vector at most. Note that some widely-used products are related to tensor product: scalar product ($l_1 = 0$, $l_2 = 1$, $l = 1$), dot product ($l_1 = 1$, $l_2 = 1$, $l = 0$) and cross product ($l_1 = 1$, $l_2 = 1$, $l = 1$). However, for each element with $l > 0$, there are multi mathematical operation for the connection with weights. The relation between number of operations and degree is quadratic. Thus, as degree increases, the amount of computation increases significantly, making calculation of the CG tensor product slow for higher order irreps. This statement can be proven by the implementation of e3nn (o3.FullyConnectedTensorProduct).

### A.2.2 LEARNABLE PARAMETERS IN TENSOR PRODUCT

We utilize the e3nn library (Geiger et al., 2022) to implement the corresponding tensor product. It is crucial to emphasize that the formulation of CG tensor product is devoid of any learnable parameters, as CG coefficients remain constant. In the context of e3nn, learnable parameters are introduced into each path, represented as $w(\mathbf{u}^{l_1} \otimes \mathbf{v}^{l_2})$. Importantly, these learnable parameters will not destory the equivariance of each path. However, they are limited in capturing directional information. In equivariant models, the original CG tensor product primarily captures directional information. We have previously mentioned our replacement of the CG tensor product with learnable modules. It is worth noting that our focus lies on the CG coefficients rather than the learnable parameters in the e3nn implementation.

### A.2.3 GATE ACTIVATION AND NORMALIZATION

The gate activation and normalization used in HDGNN are implement by e3nn code framework.

**Gate Activation.** In equivariant models, the gate activation combines two sets of group representations. The first set consists of scalar irreps ($l = 0$), which are passed through standard activation functions such as sigmoid, ReLU and SiLU. The second set comprises higher-order irreps (($l > 0$)), which are multiplied by an additional set of scalar irreps that are introduced solely for the purpose of the activation layer. These scalar irreps are also passed through activation functions.

The gate activation allows for the controlled integration of different types of irreps in the network. The scalar irreps capture global and local patterns, while the higher-order irreps capture more complex relationships and interactions. By combining these irreps in a gate-like manner, the gate activation enables the model to selectively amplify or suppress information flow based on the importance of different irreps for a given task.

**Normalization.** Normalization is a technique commonly used in neural networks to normalize the activations within each layer. It helps stabilize and accelerate the training process by reducing the internal covariate shift, which refers to the change in the distribution of layer inputs during training.

The normalization process involves computing the mean and variance across the channels. In equivariant normalization, the variance is computed using the root mean square value of the L2-norm of each type-$l$ vector. Additionally, this normalization removes the mean term. The normalized activations are then passed through a learnable affine transformation without a learnable bias, which enables the network to adjust the mean and variance based on the specific task requirements.

In our model, normalization provides an additional advantage of calibrating radial features from different representations. By incorporating layer normalization, the representation produced by equation 10 becomes more effective, especially for high-order terms.

### A.3 Relationship Between Expressive Power and Equivariant Operations

In (Dym & Maron, 2021), Theorem 2 establishes the universality of equivariant networks based on the TFN structure:

**Theorem.** For all $n \in \mathbb{N}, \mathbf{l}_T \in \mathbb{N}_+^*$,

1. For $D \in \mathbb{N}_+$, every G-equivariant polynomial $p : \mathbb{R}^{3 \times n} \to W_{\mathbf{1}_T}^n$ of degree $D$ is in $F_{C(D),D}^{TFN}$.
2. Every continuous G-equivariant function can be approximated uniformly on compact sets by functions in $\cup_{D \in \mathbb{N}_+} F_{C(D),D}^{TFN}$.

Here, $n$ represents the number of input points, $\mathbf{l}_T$ represents the degree of the approximated G-equivariant function, $C$ represents the number of channels, and $D$ represents the degree of the TFN (Tensor Field Network) structure, which is equivalent to the term $l$ used in our HDGNN. The TFN structure consists of two layers, including convolution and self-interaction. Self-interaction involves equivariant linear functions. The convolution operation calculates the CG tensor product between different irreducible representations, which is a fundamental operation for transforming directional information. Most equivariant models based on group representations use a similar approach (CG tensor product) to capture directional features. Therefore, the theorem mentioned above also applies to building blocks based on CG tensor products, such as SEGNN (Brandstetter et al., 2021) and Equiformer (Liao & Smidt, 2023).

The above theorem demonstrates that achieving an infinite degree in practice is not feasible. However, equivariant models based on group representations can enhance their expressive power by increasing the number of maximal degrees (Dym & Maron, 2021). In their evaluation of expressive power, as presented in (Joshi et al., 2023), the authors utilize the GWL (geometric Weisfeiler-Leman) graph isomorphism test. In Table 2 of their work, it is evident that equivariant models with a maximal degree denoted as $L$ are incapable of distinguishing $n$-fold symmetric structures when $n$ exceeds the value of $L$.

To assess the impact of non-equivariant operations on expressive capabilities, we designed several message modules: 1. Node embeddings, distances, and atomic number information of node $i$ and node $j$ are fed into a two-layer MLP ($MLP_0$); 2.Building upon $MLP_0$, we add the spherical harmonics of $\mathbf{r}_{ij}$ (with a maximum degree of $l$) into the node embeddings ($MLP_l$). In addition to these modules, we also tested the HDGNN module. Our observations from Table 6 reveal that MLPs can distinguish high-order symmetric structures through low-order irreps. However, they are limited in representing infinite degrees. For example, when $l = 1$, the model cannot distinguish the 10-fold structure. Furthermore, the message block of HDGNN exhibits similar performance characteristics. In practical applications, we cannot directly apply the $MLP_l$ model as it tends to excessively compromise equivariance, resulting in poor generalization performance on unseen data. Our HDGNN is designed to mitigate the loss of equivariance while retaining the strong representation capabilities of MLPs.

## B Model Details

### B.1 Invariant Branch

The purpose of the invariant branch is to support the feedforward propagation of the equivariant branch. It is important to emphasize that the input $\mathbf{f}_i$ of the attention module must be invariant to ensure the equivariance of the model. This assertion is validated in subsequent experiments. To tackle this issue, we introduce a strictly invariant branch that spans the entire model and employs invariant message passing. The message block and update block of the invariant branch are depicted in Figure 1.

Table 6: Rotationally symmetric structures. The MLP, when fed with low-degree spherical harmonics of $\mathbf{r}_{ij}$, can distinguish between two distinct rotated versions of high-order symmetric structures.

| Model | 2-fold | 3-fold | 5-fold | 10-fold |
|---|---|---|---|---|
| $TFN_{l=1}$ | $50.0 \pm 0.0$ | $50.0 \pm 0.0$ | $50.0 \pm 0.0$ | $50.0 \pm 0.0$ |
| $TFN_{l=2}$ | $100.0 \pm 0.0$ | $50.0 \pm 0.0$ | $50.0 \pm 0.0$ | $50.0 \pm 0.0$ |
| $TFN_{l=3}$ | $100.0 \pm 0.0$ | $100.0 \pm 0.0$ | $50.0 \pm 0.0$ | $50.0 \pm 0.0$ |
| $TFN_{l=5}$ | $100.0 \pm 0.0$ | $100.0 \pm 0.0$ | $100.0 \pm 0.0$ | $50.0 \pm 0.0$ |
| $TFN_{l=10}$ | $100.0 \pm 0.0$ | $100.0 \pm 0.0$ | $100.0 \pm 0.0$ | $100.0 \pm 0.0$ |
| $MLP_0$ | $50.0 \pm 0.0$ | $50.0 \pm 0.0$ | $50.0 \pm 0.0$ | $50.0 \pm 0.0$ |
| $MLP_{l=1}$ | $97.5 \pm 2.5$ | $50.0 \pm 0.0$ | $50.0 \pm 0.0$ | $50.0 \pm 0.0$ |
| $MLP_{l=2}$ | $100.0 \pm 0.0$ | $100.0 \pm 0.0$ | $87.5 \pm 12.5$ | $82.5 \pm 17.5$ |
| $MLP_{l=3}$ | $100.0 \pm 0.0$ | $100.0 \pm 0.0$ | $100.0 \pm 0.0$ | $100.0 \pm 0.0$ |
| $MLP_{l=5}$ | $100.0 \pm 0.0$ | $100.0 \pm 0.0$ | $100.0 \pm 0.0$ | $100.0 \pm 0.0$ |
| $MLP_{l=10}$ | $100.0 \pm 0.0$ | $100.0 \pm 0.0$ | $100.0 \pm 0.0$ | $100.0 \pm 0.0$ |
| $HDGNN_{l=5}$ | $100.0 \pm 0.0$ | $100.0 \pm 0.0$ | $100.0 \pm 0.0$ | $100.0 \pm 0.0$ |

In message block, we calculate the message used to update the invariant branch:

$$\mathbf{f}_{ij} = \sigma\left(\mathbf{W}[\mathbf{f}_i; \mathbf{f}_j; \mathbf{c}_{ij}]\right), \tag{26}$$

where $\mathbf{W}$ denotes the learnable weight matrix and $\sigma$ denotes the SiLU activation. In addition, the concatenation results $[\mathbf{f}_i; \mathbf{f}_j; \mathbf{c}_{ij}]$ are used to generate attention coefficients in equation 11 by a 2-layer MLP. The first layer of this MLP is a dimensionality-reduction layer with reduction ratio 4. The output activation of MLP is Sigmoid. Note that the outputs are channel-wsie atenntion, which means that the sizes of $\mathbf{a}_1, \mathbf{a}_2, \mathbf{a}_3, \mathbf{a}_4$ are all $1 \times C$. At last, we do channel-wise multiplication for each message, which is shown in equation 11.

In update block, The invariant branch is updated by:

$$\mathbf{f}_i^{k+1} = \sigma(\mathbf{W}\left(\sum_{j \in \mathcal{N}(i)} \mathbf{f}_{ij}^k\right) + \mathbf{f}_i^k). \tag{27}$$

The final embedding of invariant branch is still invariant, but the directional information is lacking. We use it to assist prediction in QM9 experiments.

## B.2 MESSAGE BLOCK

**Mean-Extension Tensor Product (METP).** The input of METP consists of all the type-$l$ vectors ($l \leq L$), including the type-$l$ vectors with low orders ($l \leq L'$). Each path of the tensor product in METP maintains equivariance. We can also utilize METP to generate higher-order type-$l$ vectors. However, incorporating higher-order terms significantly increases the computational burden, even after merging all the channels. An alternative approach is to output a higher-order representation with a single channel and then share this representation across all the channels of the messages.

**Randomness in rotation.** Note that the roll rotation around the vector $\vec{\mathbf{r}_{ij}}$ is not specified. In the implementation, we compute three unit vectors, the normalized vector $\vec{\mathbf{a}_{ij}}$ of $\vec{\mathbf{r}_{ij}}$, the randomly sampled vector $\vec{\mathbf{b}_{ij}}$ orthogonal to $\vec{\mathbf{a}_{ij}}$ and their cross product result $\vec{\mathbf{c}_{ij}}$. Rotation matrix produced by $[\vec{\mathbf{b}_{ij}}^T; \vec{\mathbf{c}_{ij}}^T; \vec{\mathbf{a}_{ij}}^T]$ can transform $\vec{\mathbf{r}_{ij}}$ to $[0, 0, 1]$.

**Shared MLP.** In equation 11, we employ MLPs with two layers, utilizing the SiLU activation function. The input is flattened into a vector, treating all channels equally. The output dimension of the first layer is denoted as $h$, which is a hyperparameter. The output dimension of the second layer is $(2L+1)^2 * C$, which is subsequently transformed into an irreducible representation. It is important to note that this MLP introduces non-equivariance, as it cannot accurately capture the function $f(\mathbf{x}) = \mathbf{x} \otimes \mathbf{C}$. However, the learnable MLP may still capture directional information due to the presence of the Wigner matrix $\mathbf{D}$ in the input, which can be seen as an embedding of $\vec{\mathbf{r}}_{ij}$.

**Distance block.** In HDGNN, the distance block in meassge block can encode the distance between two nodes ($\vec{\mathbf{r}}_{ij}$). First, the distance is encoded by a set of Gaussian basis functions $\mathcal{G}_k$ with means $\mu_k$ and standard deviation $\sigma$. The means are uniformly sampled from 0 to $\gamma$ Å, which is a regarded as a

hyper-parameter. The distance feature is given by:

$$d_k = \mathcal{G}_k(\|\vec{\mathbf{r}}_{ij}\| - \mu_k, \sigma). \tag{28}$$

At the final step, we concatenate $\mathbf{d}$ with the embeddings of $z_i$ and $z_j$, resulting in $\mathbf{c}_{ij}$. When using $\mathbf{c}_{ij}$ as input for the shared MLP, we first pass $\mathbf{c}_{ij}$ through a fully-connected layer to match the hidden size $h$. We then perform element-wise multiplication between the output of the fully-connected layer and the hidden layer of the MLP. It is important to note that element-wise multiplication is an approximately equivariant operation. To illustrate this, let's consider an equivariant operation where we multiply different equivariant type-$l$ vectors by a series of invariant features. The resulting vector can still be considered equivariant and can be represented as: $[c_0 \cdot \mathbf{Y}^0; c_1 \cdot \mathbf{Y}^1; c_2 \cdot \mathbf{Y}^2; \dots]$. It should be emphasized that if we were to introduce $\mathbf{c}_{ij}$ using other strategies, such as concatenation, it would likely compromise the performance of the overall model.

**Special Orders.** The input to the MLP only preserves specific orders of $x$ and $x$, namely, $m = -l, 0$, and $l$. This operation offers the advantage of reducing the computational burden without excessive performance degradation, as these three orders can be used to infer all the remaining orders. Taking the spherical harmonics in equation 17 as an example, $m = 0$ contains only $cos(\theta)$ terms, while $m = -l$ and $m = l$ contain $sin(\theta)$ terms and $\phi$. The value of $\phi$ can be inferred from the $m = -l$ and $m = l$ terms. The other orders can be inferred by combining $cos(\theta)$, $sin(\theta)$, and $\phi$.

**Equivariance of MLP structure.** In equation 8, we perform a transformation of the CG tensor product into a representation within local coordinate frames. It is crucial to acknowledge that this approach introduces randomness in the matrix $\mathbf{R}_{ij}$. However, thanks to the strict equivariance of CG tensor product, the right-hand side of Equation equation 8 remains equivariant. To elaborate, we express $\mathbf{R}_{ij}$ in terms of $\tilde{\mathbf{R}}\bar{\mathbf{R}}_{ij}$, where $\bar{\mathbf{R}}_{ij}$ denotes a well-defined rotation matrix, and $\tilde{\mathbf{R}}$ represents a non-specified rotation matrix satisfying the condition $\tilde{\mathbf{R}}[0, 0, 1] = [0, 0, 1]$. Consequently, Equation equation 8 undergoes the transformation:

$$
\begin{aligned}
\mathbf{D}^{-1}(\mathbf{R}_{ij})\big(\mathbf{D}(\mathbf{R}_{ij})\mathbf{x}_i \otimes \mathbf{S}(\vec{\mathbf{C}})\big) &= \mathbf{D}^{-1}(\tilde{\mathbf{R}}\bar{\mathbf{R}}_{ij})\big(\mathbf{D}(\tilde{\mathbf{R}}\bar{\mathbf{R}}_{ij})\mathbf{x}_i \otimes \mathbf{S}(\vec{\mathbf{C}})\big) \\
&= \mathbf{D}^{-1}(\bar{\mathbf{R}}_{ij})\mathbf{D}^{-1}(\tilde{\mathbf{R}})\big(\mathbf{D}(\tilde{\mathbf{R}})\mathbf{D}(\bar{\mathbf{R}}_{ij})\mathbf{x}_i \otimes \mathbf{S}(\vec{\mathbf{C}})\big) \\
&= \mathbf{D}^{-1}(\bar{\mathbf{R}}_{ij})\mathbf{D}^{-1}(\tilde{\mathbf{R}})\big(\mathbf{D}(\tilde{\mathbf{R}})\mathbf{D}(\bar{\mathbf{R}}_{ij})\mathbf{x}_i \otimes \mathbf{D}(\tilde{\mathbf{R}})\mathbf{S}(\vec{\mathbf{C}})\big).
\end{aligned} \tag{29}
$$

Due to the equivarance of CG tensor product shown in equation 25, we finally transform the above equation to

$$\mathbf{x}_i \otimes \mathbf{S}^L(\vec{\mathbf{r}}_{ij}) = \mathbf{D}^{-1}(\bar{\mathbf{R}}_{ij})\big(\mathbf{D}(\bar{\mathbf{R}}_{ij})\mathbf{x}_i \otimes \mathbf{S}(\vec{\mathbf{C}})\big). \tag{30}$$

The introduction of randomness through the term $\tilde{\mathbf{R}}$ is effectively mitigated, ensuring that the equivariance of equation 8 remains unaffected by such randomness. This preservation of equivariance is similarly observed in equation 9. However, when we substitute the CG tensor product with a MLP, the equivariance is compromised. It is crucial to note that an untrained MLP does not adhere to the condition $NN(\mathbf{Dx}) = \mathbf{D}NN(\mathbf{x})$. Consequently, the compensation strategy employed for the term $\tilde{\mathbf{R}}$ in equation 29 is not applicable in this context, leading to the breakdown of strict equivariance in equation 10. The loss of equivariance can be mitigated by optimizing the MLP structure through training.

**Model's Ability to Learn Equivariance and Directional Information.** It is important to highlight that we incorporate a neural learnable structure based on equation 9 rather than equation 8. Two primary reasons underscore this choice. Firstly, the efficacy of the equivariance in MLP hinges on their ability to discern SO(3)-transformations during training. Notably, equation 9 introduces an additional transformation for the MLP, thereby enhancing its capacity to learn and capture equivariance more effectively. Secondly, directly feeding $\mathbf{D}(\mathbf{R}_{ij})\mathbf{x}'$ to neural network is not effective in capturing the directional features in $\mathbf{D}(\mathbf{R}_{ij})$ because we cannot infer directional information in $\mathbf{D}(\mathbf{R}_{ij})$ from a whole $\mathbf{D}(\mathbf{R}_{ij})\mathbf{x}'$. In equation 10, we use the original embedding $\mathbf{x}$ and $(\mathbf{D}(\mathbf{R}_{ij}) - \mathbf{I})\mathbf{x}$ as the inputs of neural network. This method can easily infer $\mathbf{D}(\mathbf{R}_{ij})$.

**Complexity.** The complexity of the message block is approximately $O(CL_{irreps}L'_{irreps} + CL'_{irreps}h)$, where $C$ is the number of channels, $h$ is the hidden number of MLP. $L_{irreps} = (L+1)^2$ and $L'_{irreps} = 3L + 1$, where $L$ is the maximal degree.

## B.3 Update Block

To understand our construction, we first present the convolution theorem. For two functions $g(x)$ and $h(x)$ in the time domain, their Fourier transforms are as follows.

$$
\begin{aligned}
G(s) &\triangleq \mathcal{F}\{g\}(s) = \int_{-\infty}^{\infty} g(x)e^{-i2\pi s x}dx, \quad s \in \mathbb{R} \\
H(s) &\triangleq \mathcal{F}\{h\}(s) = \int_{-\infty}^{\infty} h(x)e^{-i2\pi s x}dx, \quad s \in \mathbb{R}.
\end{aligned}
\tag{31}
$$

Based on convolution theorem, we know

$$
\mathcal{F}\{g \cdot h\} = \mathcal{F}\{g\} * \mathcal{F}\{h\},
\tag{32}
$$

where $*$ denotes the convolution. The convolution theorem is also suitable for correlation $\star$. Inspired by convolution theorem, we can convert the operation closed to convolution in the frequency domain to the point-wise operation in the time domain. Note that correlation is negative convolution. To keep the same order of correlation in the frequency domain in equation 31, we can use $g(-x)$ and $h(-x)$ in the time domain.

In our update block, the convolution $*$ and correlation $\star$ in steerable space are represented by

$$
\mathbf{m}_1 * \mathbf{m}_2 = \sum_{dl} \mathbf{m}_1^{dl}\mathbf{m}_2^{l-dl}, \mathbf{m}_1 \star \mathbf{m}_2 = \sum_{dl} \mathbf{m}_1^{dl}\mathbf{m}_2^{dl-l},
\tag{33}
$$

where convolution and correlation correspond to paths $l_1 + l_2 = l$ and $|l_1 - l_2| = l$, respectively. These paths are not zeros due to the CG rule equation 5. For instance, assuming $L = 4$, and there are two irreps $m_1$ and $m_2$. $m = m_1 \otimes m_2$ contain type-0, type-1, $\cdots$, type-4 vectors. We use (type-$l_1$, type-$l_2$) to denote a path based on $m_1$ and $m_2$. Taking type-2 vector of $m$ as an example, it is the aggregation of paths (1, 1), (0, 2), (2, 0), (1, 3), (3,1), (4,2), and (2,4). However, convolution only contain (1, 1), (0, 2), and (2, 0) paths. This mehtod will weaken the equivariant interactions on time domain. We introduce correlation whose corresponding paths are (0, 2), (2, 0), (1,3), (3,1), (4,2), and (2,4), including extra paths. Therefore, this combination of convolution and correlation enables us to reduce the loss caused by FFT, as demonstrated by our ablation studies 11.

**Structure of MLP.** The size of the time domain signal is $S \times C$, where $S$ represents the number of samples on the unit sphere $S^2$. We apply a 2-layer MLP with SiLU activation to process the channel axis of the time domain signals. It is important to note that this MLP is shared across all sampling points. After passing through the MLP, the resulting output, denoted as $\mathbf{o}$, has a size of $S \times C'$.

Subsequently, we employ a squeeze-and-excitation structure to recalibrate the channel-wise features of $\mathbf{o}$. Specifically, we first perform global average pooling on $\mathbf{o}$, resulting in a vector of size $C'$. This vector is then fed into a 2-layer MLP with dimensionality reduction, yielding another vector of size $C'$. This vector is multiplied element-wise with the channels of $\mathbf{o}$. Finally, we use a FC layer without activation to transform the number of channels to $C$.

**Complexity.** The number of grids is denoted as $sample = (2 * (L + 1))^2$. The complexity of the update block part is approximately $O(sample * C + sample * C^2)$, where the latter term represents the point-wise MLP layers. In an equivariant neural network, the most time-consuming part is the CG tensor product. The complexity of the tensor product between two channels is $O((L^6)$. If we directly perform CG tensor product for every two channels in the frequency domain, complexity is $O((L^6 * C^2)$. This approach does not even include interactions within more than two channels. Therefore, employing the Fourier transform actually help to capture interactions across all channels because the complexity of point-wise operation in the time domain is low.

## C Supplementary Related Works

There are a bunch of works related to message passing neural networks, equivariant neural networks and computational chemistry. We have listed equivariant neural networks closely related to ours in Section 4. Here, we discuss other molecular models. Besides, we discuss some methods to relax equivariance.

**Molecular models.** Gilmer et al. (2017) propose a message passing neural network that can effectively describe the interactions between atoms and model the chemical properties of molecules. However, this network just considers the permutation invariance and omits the rotational invariance. Cohen et al. (2018) design spherical CNNs to analyze spherical images where they propose a definition of the

Table 7: Hyper-parameters for OC20 (IS2RE) dataset.

| Hyper-parameters | Value or description |
|---|---|
| Learning rate scheduling | $\times 0.3$ at 10, 14, 16, 18 epochs |
| Warmup steps | 100 |
| Maximum learning rate | $4 \times 10^4$ |
| Batch size | 56 |
| Number of epochs | 20 |
| Weight decay | $0.5 \times 10^3$ |
| Cutoff radius Å | 8 |
| Hidden sizes of MLPs | 512 |
| $K$ | 16 |
| $L$ | 6 |
| $L'$ | 2 |
| $\lambda$ | 0 |
| max number of neighbors | 40 |

Table 8: Hyper-parameters for QM9 dataset.

| Hyper-parameters | Value or description |
|---|---|
| Learning rate scheduling | Cosine learning rate with linear warmup |
| Warmup epochs | 5 |
| Maximum learning rate | $1.5 \times 10^4$ |
| Batch size | 128 |
| Number of epochs | 400 |
| Weight decay | $0.5 \times 10^3$ |
| Cutoff radius Å | 5 |
| Hidden sizes of MLPs | 512 |
| $K$ | 12 |
| $L$ | 4 |
| $L'$ | 3 |
| $\lambda$ | 0.05 |

spherical cross-correlation that is rotation-equivariant. To efficiently compute the cross-correlation, they resort to the fast fourier transformation. The nonlinearity of the networks stems from activation function over spatial signals. Schütt et al. (2018); Lubbers et al. (2018) make use of the interatomic distances and atomic property as the inputs so that the output is invariant to the rotation. Gasteiger et al. (2020); Liu et al. (2021) expand on using pairwise interactions to include angular terms, while the representation of nodes remains rotationally invariant, as oppose to the steerable vectors in this paper. PaiNN (Schütt et al., 2021) considers the equivariant embedding beyond the invariant one, while they just make use of $l = 1$ type vector.

**Relaxing equivariance.** Equivariance is a pervasive property in real-world scenarios. However, strict equivariant models may not always be effective due to noisy or equivariance-breaking features (Wang et al., 2022). Here, we list some works that investigate the relaxing equivariance, although their motivations and tasks are different from ours. The work in (van der Ouderaa et al., 2022; Romero & Lohit, 2022) introduce non-equivariance in group convolutional networks (Cohen & Welling, 2016) and learn layer-wise levels of equivariance from data. However, these methods are designed for convolutional neural networks (CNNs) and are not easily applicable to the feedforward architecture of GNNs. In contrast, the work (Finzi et al., 2021) proposes a method that relaxes equivariance using residual pathway. While our approach also utilizes the residual pathway, our strategy for achieving soft equivariance differs significantly. Moreover, these existing methods primarily focus on adjusting the weight distribution, whereas our approach centers on transforming the input to alleviate the equivariant constraints imposed on the learnable modules.

Table 9: Ablation studies for normalization and invariant branch.

| Model | ID Energy MAE (meV) ↓ |
|---|---|
| Baseline | 554 |
| (No norm) | 570 |
| (Approximately invariant branch) | 635 |

Table 10: Ablation studies for structure of MLP.

| Model | ID Energy MAE (meV) ↓ |
|---|---|
| Baseline | 554 |
| Var 1 | 551 |
| Var 2 | 562 |
| Var 3 | 550 |

# D  DETAILS OF EXPERIMENTS

## D.1  IMPLEMENTATION DETAILS

Here, we provide some commonly used configurations that were employed in our comparative experiments. In HDGNN, the channel size of the embedding is set to 128. The sampling rate of the distance block is 128. For the FFT sampling in the update block, we use a sampling size of $(2 \times L + 1) \times (2 \times L + 2)$. The length of the invariant feature $\mathbf{f}_i$ is set to 256.

During training, we utilize the AdamW optimizer and apply the L1 loss for all experiments. The HDGNN model is implemented using PyTorch, and the transformations on the sphere are implemented using the e3nn library.

**OC20.** For our experiments on the OC20 dataset, we follow the official PyTorch benchmark. The hyperparameters of HDGNN used in the OC20 experiments are summarized in Table 7. In the comparative experiments, we evaluate HDGNN on 8 NVIDIA A100 GPUs. In the ablation studies, we evaluate HDGNN on 4 NVIDIA V100 GPUs.

**QM9.** We adopt the same data partition as TorchMD-NET (Thölke & Fabritiis, 2022) for our experiments on the QM9 dataset. The hyperparameters of HDGNN used in the QM9 experiments are summarized in Table 8. In the QM9 experiments, we set a constant value of $cons = 0.1$ for the attention coefficient in equation 11 to manually adjust the ratio of non-equivariant operations. This constant is multiplied with $a_1$, $a_2$, and $a_3$.

## D.2  HYPER-PARAMETERS OF BASELINES

In Table 1, the results of baselines is from (Zitnick et al., 2022). We follow the Open Catalyst Project framework [2] to construct our experiment. In the QM9 experiments, for most of baselines we directly use the results from their paper.

# E  SUPPLEMENTARY EXPERIMENT

In this sectin, we construct ablation experiments to investigate HDGNN comprehensively.

## E.1  NORMALIZATION AND INVARIANT BRANCH

We conducted an evaluation to assess the impact of normalization and the invariant branch on HDGNN. Both components play a crucial role in facilitating the computation of equivariant messages. In HDGNN, equation 10 is based on the property $(\mathbf{u} + \mathbf{v}) \otimes \mathbf{b} = \mathbf{u} \otimes \mathbf{b} + \mathbf{v} \otimes \mathbf{b}$, but larger radial values can disrupt this equation during implementation. To address this, we introduce normalization to control the radial size. The "No norm" setting refers to the removal of normalization.

---

[2]https://github.com/Open-Catalyst-Project/ocp

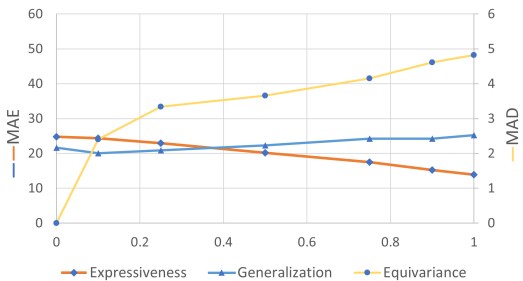

Figure 2: We show the trendency of expressiveness, equivariance and generalization. The x-axis is the ratio of non-equivariant operation. When we increase the ratio, the equivariance (measured by the negative test MAD) of the whole network decreases as expected. The generalization (measured by the negative test MAE) increases first and then decreases; the expressiveness (measured by the negative training MAE) always increases. We use 1/1000 of the original unit to represent the results.

Table 11: Ablation studies for FFT in update block.

| Model | ID Energy MAE (meV) $\downarrow$ |
|---|---|
| Baseline (Both) | 554 |
| (Conv) | 563 |
| (Corr) | 566 |
| (No FFT) | 598 |

Additionally, we investigated the performance of the invariant branch, as discussed in Section 5.2. We designed a new set of experiments to test one of our arguments: "If the invariant branch fails to maintain strict invariance, the performance of HDGNN will significantly degrade." In these experiments, we updated the invariant branch with the type-0 vector of $x_i$. From the results presented in Table 9, we made the following observations: 1. Suitable layer normalization can enhance the performance of HDGNN; 2. The invariant branch alone cannot introduce approximately invariant features.

### E.2 STRUCTURE OF MLP

The structure of the MLP has a significant impact on the performance of HDGNN. In particular, we focused on the MLPs in equation 11. We constructed three variants: 1. input-1024-output; 2. input-256-output; 3. input-512-512-output. The results, presented in Table 10, demonstrate that the performance of the baseline model is close to saturation.

### E.3 UPDATE BLOCK

In these experiments, "Conv" and "Corr" represent using only convolution or correlation, respectively. "No FFT" indicates directly using $\mathbf{m}_i$ as the new embedding after aggregating the messages $\mathbf{m}_{ij}$. From Table 11, it is evident that the combination of convolution and correlation yields the best results.

### E.4 IMPACT OF HYPER-PARAMETERS

We investigate the impact of hyperparameters in this section, and the results are presented in Table 12. We observe that the number of layers has a relatively significant impact on HDGNN. As mentioned by Dym & Maron (2021), increasing the number of layers enhances the expressiveness of the model, which aligns with our experimental findings. However, we do not observe a clear correspondence between the other hyperparameters and expressiveness in our experiments.

### E.5 ATTENTION COEFFICIENTS

We manually adjust the attention coefficients on QM9 $\mu$ task where the baseline HDGNN use the setting $C = 64$, $K = 6$, $L' = 3$ and $L = 4$. The results are shown in Figure 2, where we observe that the best result require both equivariant operations and learnable modules.

Table 12: Ablation studies for hyper-parameters.

| $K$ | $C$ | $L$ | $L'$ | ID Energy MAE (meV) ↓ |
|-----|-----|-----|------|------------------------|
| 8 | 64 | 6 | 2 | 554 |
| 8 | 128 | 6 | 2 | 557 |
| 12 | 64 | 6 | 2 | 548 |
| 8 | 64 | 4 | 2 | 579 |
| 8 | 64 | 8 | 2 | 563 |
| 8 | 64 | 6 | 4 | 551 |

Table 13: Comparison of training and inference time.

| Model | Training time (hour/epoch) | Inference time (minute/epoch) |
|-------|----------------------------|-------------------------------|
| SCN | 4.5 | 5.7 |
| HDGNN | 4.2 | 5.5 |

## E.6 TRAINING AND INFERENCE TIME

In this section, we reproduced SCN based on the official code and compared the training and inference time. The SCN we used is the SOTA model shown in Table 1. All the experiments are conducted on 8 NVIDIA A100 with batch size 56. The results are shown in Table 13.