# OpenReview forum: "Hybrid Directional Graph Neural Network for Molecules"
_ICLR.cc/2024/Conference — ICLR 2024 spotlight_

### Official Review · Reviewer_Liwe · 2023-10-18

**Soundness:** 2 fair
**Presentation:** 3 good
**Contribution:** 2 fair
**Rating:** 6
**Confidence:** 2

**Summary:**

– This paper presents a novel approach to an equivariant graph encoder for classification. The goal of the paper is to enhance expressiveness by relaxing the equivariance constraint, following recent success of works including (Dym and Maron 2021).

– The main issue is that the representation group degree $l$, which gives higher expressive power, leads to increased computation costs in the CG product.

**Strengths:**

– Investigating an hybrid approach to improve the tradeoff between equivariance and expressive power seems a fruitful direction.

**Weaknesses:**

In the abstract you claim to: demonstrating its state-of-the-art performance, however,

– Table 1: Your method does improve the results only in 2 out of 4 metrics.

– Table 2 : Your method improves previous work on 2 out of 11 tasks.

**Questions:**

– Since you are claiming that your architecture: “can enhance the expressive power suffering from the limitations of finite low-degree group representation” should this be also shown in a proposition or theorem, to complement your ablation study?

– Could you expand experiments on other recent GNN benchmarks, follow Equiformer or the other works that you are comparing to.

---

> ### Author Response · Authors · 2023-11-16
> **Responses to reviewer**
>
> Thank you for your valuable reviews. If you have any additional comments or concerns, please do not hesitate to let us know.
>
> # Responses to weekness.
> We clarify that ID, OOD Ads, OOD Cat, OOD Both are four tasks in OC20 (different data), not metrics. Our original sentence in abstract was that " demonstrating its state-of-the-art performance on several tasks". We did not say that we achieved SOTA results on all tasks. The average scores in Table 1 can comprehensively evaluate the models. In Table 1, **we achieve the best average scores and achieve 4.4% improvement. As for the subtasks, although our method did not achieve the best results on the ID and OOD Cat tasks, they both achieved the second best results and were very close to the first place.** Note that OC20 is a representative data because it represents a large swath of chemical space. In QM9 tasks, we also achieved second place in multiple indicators, very close to first place.
>
> # Responses to question 1.
> We discuss relevant theorem and experiments in Appendix A.3. Our discussion is based on two theoretical works [1] [2]. **The theorem 2 in [1] asserts that any equivariant function can be approximated by a CG tensor product with infinite degree, though practical constraints limit it to a finite low degree.** Therefore, some complex equivariant pattern can not be fitted by CG produt with low degree (please refer to Table 2 in [2]). Concurrently, **based on the approximation properties of MLP, we recognize MLP's capability to approximate any continuous function, which includes equivariant functions (please refer to figure 1 in [1]).** Our intention is to leverage MLP to overcome the limitations of CG tensor products with low degrees. **Table 6 further illustrates MLP's proficiency in learning higher-degree patterns when embedding is in a low-degree representation.** Due to space constraints and the non-original nature of the aforementioned theorem, we provide a more detailed discussion in the Appendix.
>
> [1] On the Universality of Rotation Equivariant Point Cloud Networks. Nadav Dym, Haggai Maron. ICLR 2021
>
> [2] On the Expressive Power of Geometric Graph Neural Networks. Chaitanya K. Joshi, Cristian Bodnar, Simon V. Mathis, Taco Cohen, Pietro Li.
>
> # Responses to question 2.
> Firstly, we did not assess the MD17 dataset. Given its small size, it poses challenges for HDGNN to learn the correct equivariance (other methods use inherent equivariance, expect for SCN). **Our work is designed to enhance expressive power by relaxing equivariance, enabling networks to comprehend intricate physical interactions between atoms. Consequently, HDGNN is expected to have good performance across a large swath of chemical space. OC20 stands out as representative due to its substantial size (O(100M)) and diversity (55 elements).** Additionally, Equiformer presented some results involving data augmentation on OC20 experiments, such as noisy nodes and IS2RS. This may be attributed to Equiformer's competitive nature (OC20 challenge), where the authors tend to achieve optimal MAE. In contrast, our study focuses on only comparing the performance of pure models. The basic training environment (without augmentation) can provide an accurate evaluation.
>
> Since part of the equivariance of HDGNN comes from learning, **SO(3)-transformation augmentation of MD17 may allow HDGNN to have a good performance. But this will lead to unfair comparisons (the results of other methods use the original dataset).** In addition, we are sorry that we cannot complete the data augmentation experiments on OD20 during the rebuttal time due to limited resources (OC20 experiments take up a lot of resources). We will publish these results in the public community in the future. **Nevertheless, outcomes derived from data augmentation may not precisely represent the true performance of the model, as it remains unclear which specific component contributes to the observed improvement. We believe that the pure model comparisons in our Table 1 and Table 2 can provide a significant evaluation for HDGNN.**

---

### Official Review · Reviewer_yQB9 · 2023-10-26

**Soundness:** 3 good
**Presentation:** 3 good
**Contribution:** 3 good
**Rating:** 8
**Confidence:** 2

**Summary:**

The paper proposes the Hybrid Directional Graph Neural Network (HDGNN), a message passing graph neural network designed to enhance expression power by relaxing equivariance. the experiments demonstrate promising result, and the ablation study shows the relevant module is useful.

**Strengths:**

1. the paper is well written and easy to follow
2. experiment looks good

**Weaknesses:**

No

**Questions:**

No

---

> ### Author Response · Authors · 2023-11-16
> **Response to reviewer**
>
> We appreciate your acknowledgment of our work. If you have any additional comments or concerns, please do not hesitate to let us know.

---

### Official Review · Reviewer_rs3C · 2023-11-06

**Soundness:** 2 fair
**Presentation:** 3 good
**Contribution:** 3 good
**Rating:** 8
**Confidence:** 4

**Summary:**

The paper describes an hybrid approach to equivariant graph neural networks. It is hybrid in the sense that the networks contain operations that are not fully equivariant together with those that are. The motivation is that equivariance could be too strong of a constraint in many applications. Another motivation is that the Clebsch-Gordan tensor product framework is demanding, and one could by a trick one could convert the steerable features to a global basis via the Wigner-D matrices, apply an MLP, than map back via the Wigner-D matrices. The authors achieve strong results on QM9, state of the art on various metrics for the open catalyst challenge, and perform ablation studies to investigate the effect of the breaking full equivariance.

*[update after the firs response I increased my score from 3 to 5]*

*[update after a second round of discussion I raised my score from 5 to 6]*

*[given the final modifications/clarifications I raised my score from 6 to 8, the authors effectively addressed all my initial concerns and I see no reason to reject the paper]*

**Strengths:**

1. The paper obtains excellent results on various benchmarks
2. The theoretical preliminaries part, and most of the rest of the paper, is well written
3. The paper addresses a relevant topic; it investigating whether the sometimes restrictive inductive bias of equivariance can be relaxed

**Weaknesses:**

1. A lot of focus is on breaking of equivariance, or relaxing this constraint. There is one element that I find very important which is not discussed however:

When converting the features to a global reference frame via the Wigner-D matrix (Equation 9), the rotation is computed from the direction: "$R_{ij}$ denotes the 3D rotation matrix that transforms  $\vec{r}_{ij}$ to $[0,0,1]$".

However, this is ill-defined as there is one degree of freedom left in the rotation matrix as any $R_{ij} R_{\alpha}$ with $R_{\alpha}$ any rotation around the z-axis results in the same mapping from $\vec{r}_{ij}$ to $[0,0,1]$.

Thus, the equality in equation 9 is not true, and this is an error in the paper. Iis this is a dellibarate error, because you want to break equivariance? Eitherway, it should fixed or commented on.

It seems that this type of trick as seems to be adopted from [Zitnick et al. 2022], although I haven't check if the same type of error is present in that paper. However, working with local reference frames was also presented in [Kofinas et al. 2021]. See section 3.2 for their discussion on the ill-defined rotation. The claim "... was first presented in Zitnick et al" might therefore have to be also updated/nuanced.

[Kofinas et al. 2021] Kofinas, M., Nagaraja, N., & Gavves, E. (2021). Roto-translated local coordinate frames for interacting dynamical systems. Advances in Neural Information Processing Systems, 34, 6417-6429.

2. A second weakness of the paper is the complexity of the model (figure 1) with all sorts of interactions and blocks. It is great that excellent results are obtained with the model, but is it really worth the engineering effort? In my opinion, the contribution of great results is greatly diminished if it comes at the cost of such an intricate (over engineered?) network design. Then, what remains is the ablation that shows that relaxing equivariance could be benefficial. This is a message that contains scientific value, in my opinion, however, this point I think is not thoroughly discussed/analyzed.

3. The main innovation of this paper, which is the partial equivariance is not clearly explained, in my opinion. Apart from the error with respect to ill-defined rotation matrix estimation, the explanaition of why one type of processes breaks equivariance and the other doesn't (Eq 11) could be better explained.

4. In general, I think the paper could still benefit form being more precise on the the purpose of all included experiments (what precisely do they test for?)

Detailed comments:
1. In the introduction it is mentioned "While equivariant neural networks exhibit appropriate inductive biases ... leading to inefficient learning of interactions between atoms" I do not agree with (or understand) this claim at all. Precisely these domains require equivariance: it is demanded by the physics. Any operation that is non-equivariant is bound to fail, unless it *learns to be equivariant*. The property of equivariance is either way essential and thus imposing equivariance should improve efficiency in learning. Perhaps the authors mean something like that it might be more efficient to start-off unconstraint and let the network *learn equivariance* (though the task or augmentation). The paper is not clear on this, it is often suggested that one wants to break equivariance because it might not be the right inductive bias. Eitherway, I really do not understand what is meant with the statement, and I think it is wrong. Please give a counter example where the predicted properties do not require equivariance/invariance, or update the sentence.

2. In 2.4, the analysis of expressive power is I think more important than the benchmark results but hidden in the appendix. Could the results be somehow briefly discussed in the main body, instead of an appendix that is not included in the main pdf?

3. Above equiation 6 in the neighborhood definition please use $\mid$ (\mid) to leave some space between the left and right of | . Also equation 6 reads better if you include some white space after the comma.

4. Please include a discussion or more details on the estimation of $R_{ij}$ from $\vec{r}_{ij}$, as mentioned in my main comments.

5. Above equation 10: "However, the MLP structure introduces non-equivariance" Please explain this. I do not see why it would break equivariance. In other words, if equation 9 were true (if the rotation estimate were to be well-defined), then applying an MLP on $x_i'$ and then rotate it back afterwards would be fine, right? It would only break equivariance because of the ill-defined rotation estimation, but that is separate from the use of an MLP. So I wonder what is the issue here really? Could you explain this?

6. Same sentence continues with "... rotation matrix introduces randomness". Is this because of the free rotation angle, or what is meant here?

7. Equations 10 and 11 could benefit from more explanation. Why this decomposition? What problem does it solve? Is it to split the processing into an equivariant and non-equivariant part, through NN(x_i) or NN(x_i'')? Or what is the idea behind it.

8. In equation 12, what are the $a_{ij}$, and how are they computed?

9. Equations 13 and 14 describe a Fourier based activation function right? Where as "activation function" an MLP is used. I do like this approach but I think the presentation could be simplified a bit.

10. After equation 13: "we concatenate all the elements..." why? It seems very ad-hoc, but perhaps there is a good explanation for doing this. Could you explain why this is done?

11. Next, "Here, we extend the point-wise operation to two points by incorporating both convolution and correlation" What does that mean? Convolution and correlation are one and the same thing, up to a flip of the kernel. This sentence makes no sense to me, please clarify what is meant by it.

12. In related works: "Such structure is limited to capture fine-grained geometric features since the learning weights are only in the radial representation which is a scalar in each filter". *This statement overlooks the fact that also the spherical harmonics are used*. The radial basis, in combination with the spherical harmonic basis makes that the CG operations describe full group convolutions with kernels that can in principle approximate any pattern. They are only limited by the band-limit L. **Saying that the use of the radial scalar is limiting is wrong and misleading because it overlooks the use of Y**. See e.g. for more details on the expresiveness of CG based networks and the band-limit L:

Weiler, M., Geiger, M., Welling, M., Boomsma, W., & Cohen, T. S. (2018). 3d steerable cnns: Learning rotationally equivariant features in volumetric data. Advances in Neural Information Processing Systems, 31.

or

Cesa, G., Lang, L., & Weiler, M. (2021, October). A program to build E (N)-equivariant steerable CNNs. In International Conference on Learning Representations.

13. "In contrast, our method ... striking a balance between equivariance and expressiveness". Again, I find a critical discussion on this balance missing. *All the addressed tasks demand equivariance*. The discussion should focus therefore on how certain types of equivariant networks limit expresivity. In CG approach this limit is I suppose the band-limit L. In your approach there is no inherent band-limit because of the non-equivariant MLP approach (though I'm not sure how this is proved), but equivariance should be learned somehow. Do I understand this correctly?

14. Below table 1: " This highlights the ability of HDGNN to capture more general physical properties in real world scenarios" I do not see this claim substantiated. The only thing I see from the table is that the method outperforms several others (not always), but this does not imply a better capability to "capture more general physical properties". What are these "more general properties". There are so many differences between your architecture and those in the table. In my interpretation, HDGNN might just be better engineered to solve the task but does not give any more insights than that.

15. What are the ODD average scores between parenthesis in text (GDGNN: 564 vs SCN/Equiformer: 592/613). I cannot find these in the table (sorry if I overlook them somehow)

16. "The QM9 dataset ... susceptible to overfitting" I do not see why this is an issue. Did you indeed observe overfitting in your learning curves? If you want to test these kind of overfitting issues, perhaps the revised rMD17 would be more appropriate. In fact, I think rMD17 might be the best dataset to test your method, because the task in rRMD17 (interatomic potential energy prediction, for a single molecule given different conformations) is purely driven by geometry and should be purely invariant.

17. "In contrast, our approach merges inherent equivariance with learned equivariance," What mechanism induces learned equivariance? This is not discussed in the paper.

18. "While HDGNN has shown superior performance compared to strictly equivariant models" did it? It seems sometimes it does, sometimes it doesn't. The claim should be nuanced.

19. To be honest, I do not follow the paragraphs after tables 3,4. They should be improved in clarity. What research questions are being addressed here? Please be precise on the purpose of all experiments in the paper (what precisely do they test for)

20. On several occasions the paper motivates the equivariance relaxation form the efficiency perspective. But what about actual speed efficiency? I see no comparison in the paper and it would be nice to see if something is gained on that level.

**Questions:**

See questions in comments above. Thank you for your great efforts, it is an interesting paper, and thanks for considering my comments.

---

> ### Author Response · Authors · 2023-11-16
> **Responses to weekness 1-3 in main comments**
>
> Thank you for your valuable reviews. We have incorporated many of your suggestions to make the article clearer and more rigorous (Modified parts are highlighted in red). To streamline your review process, we offer guidance on locating specific responses. **Answers concerning randomness can be found in response 1 of the main comments, and responses 4-6 of the detailed comments. Insights into breaking equivariance operations are addressed in response 3 of the main comments, along with responses 5 and 7 of the detailed comments. Clarification on the Fourier transform on the sphere is provided in responses 9-11 of the detailed comments.** If you have any additional comments or concerns, please do not hesitate to let us know.
>
> # Responses to weekness 1 in main comments.
> While acknowledging the existence of multiple rotation transformations $\mathbf{R} _{ij}$ capable of rotating $\vec{\mathbf{r}} _{ij} $ to [0,0,1], **the equality in Equation 9 remains valid**. In detail, we use two rotation parameters, namely $\theta$ and $\varphi$ (changes in the polar and azimuth angles), to denote $\mathbf{R} _{ij}$. $\varphi$ is random when $\mathbf{R} _{ij}$ rotate $\vec{\mathbf{r}} _{ij}$ to [0,0,1]. We randomly select one parameter configuration and consistently apply it throughout the subsequent processes. With the choosen $\mathbf{R} _{ij}$, we get $\mathbf{D}^{-1}(\mathbf{R} _{ij})\mathbf{D}(\mathbf{R} _{ij})=\mathbf{I}$. Consequently, we initially convert the left side of Equation 9 to $(\mathbf{D}^{-1}(\mathbf{R} _{ij})\mathbf{D}(\mathbf{R} _{ij})\mathbf{x} _{i}) \otimes (\mathbf{D}^{-1}(\mathbf{R} _{ij})\mathbf{D}(\mathbf{R} _{ij}) \mathbf{S}(\vec{\mathbf{r}} _{ij}))$. Based on the definition of $\mathbf{x}' _{i}$ and the Wigner-D matrix, this formula transforms into $(\mathbf{D}^{-1}(\mathbf{R} _{ij})\mathbf{x}' _{i}) \otimes \mathbf{D}^{-1}(\mathbf{R} _{ij})\mathbf{S}(\mathbf{C})$. Additionally. The CG product is equivariant for all rotations (refer to Equation 26 in the Appendix). Consequently, the above formula can be transformed into the right side of Equation 9.
>
> In summary, equation 9 can not be affected by randomness because of the equivariance of CG product. The $\mathbf{D}^{-1}$ is the transpose of $\mathbf{D}$ in the implementation (They are based on the same $\mathbf{R}_{ij}$). In this case, **the outputs of the right side of equation 9 is the same when we choose different $\mathbf{R}_{ij}$. We deliberately break equivariance in Equation 11**, incorporating the non-equivariant MLP (You can refer to Responses to 7-th detailed comment).
>
> Reading reponse to the 5-th detailed comment together may better address your concerns.
>
> Thank you for your valuable suggestions. We updated two parts in the new version: **1.We provide a more rigorous reference and description to the local reference frames. 2.We will add the disscusion that Equation 9 will not influence by randomness.**
>
>
> # Responses to weekness 2 in main comments.
>
> Figure 1 may appear intricate for two main reasons:
>
> 1. We show most of the details to improve its reproducibility. Analogues can be found in other MPNNs, such as the Figure 2 in GemNet [1] and Figure 1 in Equiformer [2]. **While the design of many MPNNs may seem complex, it's important to note that most of modules are fixed. Therefore, we can bypass the complex hyper-parameter design in engineering and easily apply models to various molecular tasks.** Our approach is similar. In our ablation (Appendix E.4), we show that HDGNN mainly replies on four hyper-parameters ($K$, $C$, $L$, $L'$).
>
> 2. **Some operations depicted in our Figure 1 share the concise high-level scientific ideas.** For instance, the neural network paths in Multi-TP share the similar non-equivariant design. **The modules in Figure 1 can be summarized to two scientific ideas: 1.the design of approximately equivariant modules; 2.the combination between equivariant modules and non-equivariant modules.** We discuss the motivation of ideas in introduction and section 2.4, the evolutions process of ideas in section 3.2 and 3.3. We have emphasized the core scientific ideas in new version and indicated their corresponding building blocks.
>
> [1] Gasteiger J, Becker F, Günnemann S. Gemnet: Universal directional graph neural networks for molecules. Neurips 2021.
>
> [2] Liao Y L, Smidt T. Equiformer: Equivariant graph attention transformer for 3d atomistic graph. ICLR 2023
>
> # Responses to weekness 3 in main comments.
> Thank you for the suggestion. We will address this in the new version. We clarify that **two components in Equation 11 break equivariance.** This is because the non-equivariance of MLP (You can refer to Responses 5 to detailed comments). **The equivariant component in message block is METP operation.** Note that the final message is calculate by Equation 12. We use equations 8 and 11 to introduce the equivariant and non-equivariant parts of equation 12, respectively.

---

> ### Author Response · Authors · 2023-11-16
> **Responses to weekness 4 in main comments and weekness 1-6 in detailed comments**
>
> # Responses to weekness 4 in main comments.
> **The results in Table 1 and Table 2 represent the generalization ability on validation, which reveal whether models can accurately decipher the underlying physical or chemical principles inherent in molecular models. In our ablation experiments, we systematically examine how specific sub-modules in HDGNN influence the generalization ability.** In essence, generalization ability is influenced by both expressive power and equivariance. The modules in HDGNN impact these two aspects. When equivariance is severely damaged, generalization ability will also be severely reduced ("Unrestricted" in Table 3). Additionally, insufficient expressive power also constrains generalization ability. In Table 3, we employ strictly equivariant, approximately equivariant, and unconstrained operations to compute messages within our model framework. The results confirm that approximate equivariance is able to enhance generalization. Table 4 assesses our attention method, which is designed to dynamically adjust the ratio of non-equivariant to equivariant operations ($a _{ij}$ in Equation 12). This outcome illustrates our method's endeavor to strike a balance between expressive power and equivariance for improved overall generalization. Table 5 is dedicated to validating the detail structure or design idea of the core module. A more extensive evaluation is provided in the Appendix. These experiments serve to facilitate the reproducibility and customization of HDGNN.
>
> # Responses to weekness 1 in detailed comments.
> Thank you for your review, we have updated this sentence to make a rigorous expression. We are sorry that our expression caused you misunderstanding and hope our rebuttal can clarify it. What we want to express is that the constraints in equivariant operation (sband-limit degree) can impact the model's expressive capacity, leading to inefficient learning. In this context, inefficient learning refers to the model's inability to capture accurate mathematical relationships between atoms.
>
> # Responses to weekness 2 in detailed comments.
> We have included the analysis of expressive power in the Appendix for two main reasons: 1. Given the constraints on text length, our primary aim is to maximize the introduction of our main contribution—HDGNN. While the analysis of expressive power serves as a crucial motivation, the intricate details of this aspect are complex. Consequently, we can only provide a concise overview of the core ideas. 2. The theorems related to expressive power in equivariant models are not our contributions. Excessive description in the main text may lead to misunderstanding of our contribution. Therefore, we only put the most important conclusions in the introduction and section 2.4.
>
> # Responses to weekness 3 in detailed comments.
> Thanks for your suggestion, we have updated.
>
> # Responses to weekness 4 in detailed comments.
> Thank you for your suggestion. We have updated additional details and discussions concerning $\mathbf{R} _{ij}$. In practice, we first produce normalized vector of $\vec{\mathbf{r}} _{ij}$, denoted as $\vec{\mathbf{a}}$. We then randomly sample an orthogonal vector $\vec{\mathbf{b}}$ of $\vec{\mathbf{r}} _{ij}$ on the sphere, and calculate another orthogonal vector $\vec{\mathbf{c}}$ through cross product. $\mathbf{R} _{ij} = [\vec{\mathbf{b}}^{T}; \vec{\mathbf{c}}^{T}; \vec{\mathbf{a}}^{T}]$. This approach ensures orthogonality of $\mathbf{R} _{ij}$ and is similar to produce $\mathbf{R} _{ij}$ by random $\varphi$.
>
> # Responses to weekness 5 in detailed comments.
> **MLP breaks equivariance because its functional form is non-equivariant to SO(3)-transformation.** MLP does not conform to the definition in Equation 1, when $T(g)$ denotes SO(3)-transformation. In simpler terms, **when a 3-D vector $\mathbf{x}$ (or an irreducible representation) is input into MLP, the outcome violates the equality $NN(\mathbf{R} \mathbf{x}) \not= \mathbf{D} NN(\mathbf{x})$.** Here, $\mathbf{R}$ represents any rotation and $\mathbf{D}$ represents corresponding Wigner-D matrix.
>
> For example, we assume that the input $\mathbf{x}$ is based on the spherical harmonic basis. **We can use $\mathbf{D}^{-1}(\mathbf{R} _{ij})$ to rotate it back only when $NN(\mathbf{D}(\mathbf{R} _{ij}) \mathbf{x}) = \mathbf{D}(\mathbf{R} _{ij}) NN(\mathbf{x})$.** However, an untrained MLP cannot satisfy this condition (its output is not based on the spherical harmonic basis).
>
> Randomness will amplify the impact of non-equivariance of MLP. When we replace $\mathbf{x}_i \otimes \mathbf{S}(C)$ with a untrained MLP in equation 9, different $\mathbf{R} _{ij}$ lead to different results. In this case, the equality of equation is not true, and the equivariance on the right side breaks down.
>
> # Responses to weekness 6 in detailed comments.
> Yes, the randomness is caused by free rotation angle.

---

> ### Author Response · Authors · 2023-11-16
> **Responses to weekness 7-10 in detailed comments**
>
> # Responses to weekness 7 in detailed comments.
> We first clarify the equivariance of each term and purpose of each equation. Equation 10 is strictly equivariant, and $NN(\mathbf{x})$ and $NN(\mathbf{x}'')$ in equation 11 are both non-equivariant. Equation 9-11 are expansion of Equation 7 which is based on equivariant CG tensor product. We mentioned that Equation 7 is widely used in other equivariant models (the sentence after Equation 7). However, finite degree limits expressiveness. In equation 11, we aim to replace CG product by MLP to improve expressiveness.
>
> Second, we introduce the idea of decomposition. If there is no decomposition, then Equation 9 will be approximated by $\mathbf{D}_{-1}(\mathbf{R} _{ij})NN(\mathbf{x}')$. This will cause two problems: 1. The randomness in $\mathbf{x}'$ will affect the learning of MLP. 2. We hope that MLP can capture directional features of edge. In equation 9, these features are stored in the wigner-D matrix ($\phi$ is random, but $\theta$ is well-defined). However, the input of MLP ($\mathbf{x}'=\mathbf{D}\mathbf{x}$) is a whole, and we cannot expect MLP to capture the information in $\mathbf{D}$ separately from this whole. Through the decomposition of equation 10 and equation 11, one of the branches $NN(\mathbf{x} _{i})$ is not affected by randomness. Moreover, we feed $(\mathbf{D}-\mathbf{I})\mathbf{x}$ and $\mathbf{x}$ to the MLP, making it possible to extract the directional features in $\mathbf{D}$. We discussed this decomposition in the sentences following Equation 11.
>
> We add some content here to facilitate understanding the role of MLP. We use MLP to learn the CG product process $f(\mathbf{x}) = (\mathbf{x} \otimes \mathbf{S}(C))$. Due to MLP's ability to approximate any function, it can even extend the learned function to higher-degree CG tensor products. Table 6 verified that MLP can learn features in higher-degree representations.
>
> # Responses to weekness 8 in detailed comments.
> $\mathbf{a} _{ij}$ is the attention coefficient, they are generated by invariant branches. The formula is $MLP(\mathbf{f} _i, \mathbf{f} _j, \mathbf{c} _{ij})$, which corresponds to the green module in figure 1. Thank you for pointing this out, we have added a description of it in the new version. The attention coefficients are used to automatically adjust the weights of non-equivariant and equivariant operations. Due to limited space, the details of invariant branch $\mathbf{f}$ are put in Appendix B.1.
>
> # Responses to weekness 9 in detailed comments.
> Equations 13 and 14 represent the Fourier transform on the sphere. Spherical harmonic can be regarded as frequency representation of Fourier transform on a sphere (Appendix A.1.2). The CG tensor product can be regarded as a special convolution process. Based on the convolution theorem, **we can use point-wise operations in the time domain to approximate the convolution process in the frequency domain.** Therefore, in equation 13, we first convert the signal in the frequency domain to the time domain, and then perform a point-wise MLP in the time domain. Finally, we convert it to the frequency domain through equation 14. We do this to reduce the amount of computation. **The purpose of the update block is to calculate the interaction between different channels of each node embedding. If we reference the CG product to each channel pair, the computational cost will become very high. This approach also cannot consider the interaction between three or more channels.** Although the point-wise MLP in the time domain is only an approximation, its complexity is much smaller and interactions between mutiple channels are considered.
>
> # Responses to weekness 10 in detailed comments.
> When we perform point-wise operations in time domain, the point refers to each direction vectors $\mathbf{r}$. Taking the convolution theorem as an example, assume that node embedding in the frequency domain has only two channels, denoted by $m _1$ and $m _2$. We use $p _1$, $p _2$ to denote their signals in time domain. We can use point-wise multiplication $p _1 \circ p _2$ to approximate convolution ($\circ$ is the Hadamard product). For each point $\mathbf{r}$, the output is $p _1(\mathbf{r}) * p _2(\mathbf{r})$. Now, we have C channels and use MLP as point-wise operation. The calculation at point $\mathbf{r}$ can be expressed as $MLP([p _1(r), p _2(r), ..., p _C(r)])$. Moverover, we extend to correlation so also include $\mathbf{-r}$.

---

> ### Author Response · Authors · 2023-11-16
> **Responses to weekness 11-20 in detailed comments**
>
> # Responses to weekness 11 in detailed comments.
> We regard CG tensor product as a special convolution because some paths in CG tensor product are similar to the form of convolution. However, the paths outside of the convolution will be missing. We supplement the missing path through correlation. For example, if we calculate CG product between two irreducible representation $m_1$, $m_2$. We assume $L=4$, and $m_1$, $m_2$ and $m'=m_1 \otimes m_2$ contain type-0, type-1, $\cdots$, type-4 vectors. (type-$ l_1$, type-$l_2$) denotes a path based on $m_1$ and $m_2$. Taking type-2 vector of $m'$ as an example, it is the aggregation of paths (1, 1), (0 , 2), (2, 0), (1, 3), (3,1), (4,2), and (2,4) (refer to equation 5). However, convolution only contains (1, 1), (0, 2), and (2, 0) paths (refer to equation 32). This mehtod will weaken the equivariant interactions they learn. We introduce correlation whose corresponding paths are (0, 2), (2, 0), (1,3) , (3,1), (4,2), and (2,4), including paths that are not belong to convolution. In equation 5, convolution/correlation corresponds to $(l_{1}+l_{2}= l)/|(l_{1}-l_{2}|=l)$ paths. The conversion of correlation to the time domain is an operation between ($\mathbf{r}$) and ($\mathbf{-r}$). Therefore, we concatenate the ($\mathbf{r}$) and ($\mathbf{-r}$) of all channels in the time domain.
>
> **Although the difference between convolution and correlation is only one negative sign, they correspond to different paths of CG tensor product in the frequency domain.** We discussed this in appendix B.3. Table 11 also proves that introducing correlation can improve performance.
>
> # Responses to weekness 12 in detailed comments.
> Thanks for your reviews, we have updated this sentence.
>
> # Responses to weekness 13 in detailed comments.
> Your understanding is correct. We have updated this sentence to make a rigorous expression. This sentence is used to compare with SCN. What we want to express is that we introduce equivariance operations while learning equivariance, improving expressiveness while also preserving equivariance (the equivariance of SCN is mainly from learning). **We are glad that we have similar insights as you on the limitations of CG tensor product (band-limit degree). In our paper, this band-limit degree is represented by "finite degree". We mainly discussed in the second paragraph of the introduction and section 2.4.**
>
> # Responses to weekness 14 in detailed comments.
> Thanks for your review. we treat it as a conjecture in new version to make a rigorous expression. Note that the previous sentence of this sentence is a comparison of OOD. OOD represents unseen distribution (we briefly introduced OOD in section 5.1). **Atomic energy is related to some physical mechanisms, such as Coulomb and van der Waals forces. However, due to the properties of machine learning, what the model learns may not be the general physical mechanism, but some special mechanisms in training data**. HDGNN performs better on unseen distribution, so we infer that what it learned may be closer to the real physical mechanism.
>
> # Responses to weekness 15 in detailed comments.
> These scores ​​are the average of OOD tasks (OOD ads, OOD Cat, and OOD Both). The average scores in the Table 1 is the average of all tasks (ID, OOD ads, OOD Cat, OOD Both).
>
> # Responses to weekness 16 in detailed comments.
> We have observed overfitting in all the tasks of QM9. For example, The final training MAE is much lower than the validation MAE (in the strictly equivariant model, they are similar, you can refer to the open training log of Equiformer). The reason for overfitting is that the dataset of QM9 is relatively small, and HDGNN cannot learn the equivariance under SO(3) transformation. If we perform SO(3) transformation augmentation, this overfitting may be alleviated. However, this method will bring an unfair comparison (other methods use the original dataset). Thanks for your suggestion, we will evaluate rMD17 in the open source community in the future (sorry that the available resources are currently limited). QM9 and IS2RE are also purely invariant, and lead a comprehensive comparison to SOTA molecular models. Many advanced methods have published code and results on them.
>
> # Responses to weekness 17 in detailed comments.
> The MLP in equation 12 introduces learned equivariance. The METP introduces inherent equivariance.
>
> # Responses to weekness 18 in detailed comments.
> Thanks for your suggestion. We have updated.
>
> # Responses to weekness 19 in detailed comments.
> Please refer to our response to 4-th main comments.
>
> # Responses to weekness 20 in detailed comments.
> Thank you for your suggestion. The experiments assessing speed efficiency are presented in Table 13 in the Appendix. Furthermore, **we have added the complexity analysis of our operations in Appendix B.2 and B.3.** This update includes a comparative assessment of the efficiency between our proposed structure and the CG tensor product.

---

> ### Comment · Reviewer_rs3C · 2023-11-17
>
> I raised my score a bit (from 3 to 5) due to a lacking validity of the mentioned weakness 1 from my side. Even though the results are great, I cannot recommend accept because I am still not convinced about the motivation behind the method, it still seems like the design is flawed but somehow this flaw is turned into a feature. In particular the notion of "approximate equivariance" is not convincing, since there seems nothing equivariant about the method left after all the "approximations". I elaborate my concerns below. I highlighted four statements [**A**], [**B**], [**C**], [**D**], which I hope you could still respond to.
>
> **Regarding weakness 1**
>
> Thank you for your patience with me, and I apologize for my rushed review. My statement that equation 9 was incorrect, was incorrect. The identity is clearly true. *My listed weakness number was not valid, furthermore, I think it has been appropriately addressed in the new text.* However, stating that randomness does not affect equation 9, does not solve the following issue.
>
> **Regarding weakness 3**
>
> *Apologies for the poor latex formatting, somehow it wouldn't render the subscripts correctly*
>
> What I was concerned about were actually the next steps, after the decomposition using equation 9. My comment number 3 addresses this concern of mine, and it is still not fully resolved. To clarify my thoughts, firstly, let us validate when the operations are equivariant and when not, and check the validity of changing the original tensor product  $\mathbf{D}(\mathbf{R}_{ij})^{-1} (\mathbf{D}( \mathbf{R}{ij} ) \mathbf{x}_i) \otimes \mathbf{S}(\vec{\mathbf{C}})$ by
> $\mathbf{D}(\mathbf{R}{ij})^{-1} NN(\mathbf{D}(\mathbf{R}{ij}) \mathbf{x}_i)$.
>
>
> For now assume a well-defined unique rotation $\mathbf{R}{ij}$ such that $\mathbf{r}{ij}=\mathbf{R}{ij}^{-1} \vec{\mathbf{C}}$, as defined in the paper, then a global rotation action on the point cloud would lead to
> $$
> \mathbf{r}i \mapsto \mathbf{R} \mathbf{r}i = \mathbf{R} \mathbf{R}{ij}^{-1} \vec{\mathbf{C}} = (\mathbf{R}{ij} \mathbf{R}^{-1})^{-1} \vec{\mathbf{C}} \, ,
> $$
> and thus a global rotation leads to $\mathbf{R}{ij} \mapsto \mathbf{R}{ij} \mathbf{R}^{-1}$. If all layers are equivariant, then the corresponding transformation to feature vectors $\mathbf{x}i$ would be via $\mathbf{x}i \mapsto \mathbf{D}(\mathbf{R}) \mathbf{x}i$. The corresponding transformation to the NN layer would thus be
> $$
> \mathbf{D}(\mathbf{R}{ij} \mathbf{R}^{-1})^{-1} NN(\mathbf{D}(\mathbf{R}{ij} \mathbf{R}^{-1}) D(\mathbf{R}) \mathbf{x}i) =
> $$
> $$
> \mathbf{D}(\mathbf{R}) \mathbf{D}(\mathbf{R}{ij})^{-1} NN(\mathbf{D}(\mathbf{R}{ij} ) \mathbf{D}(\mathbf{R}^{-1}) D(\mathbf{R}) \mathbf{x}i) =
> $$
> $$
> \mathbf{D}(\mathbf{R}) \mathbf{D}(\mathbf{R}{ij})^{-1} NN(\mathbf{D}(\mathbf{R}{ij} ) \mathbf{x}i) =
> $$
> $$
> \mathbf{D}(\mathbf{R}) \mathbf{D}(\mathbf{R}{ij})^{-1} NN(\mathbf{x}i') \, ,
> $$
> and we nicely have that the output of such a transformation is still steerable via a Wigner-D matrix applied on the left, namely we have $\mathbf{D}(\mathbf{R}{ij})^{-1} NN(\mathbf{x}') \mapsto \mathbf{D}(\mathbf{R}) \mathbf{D}(\mathbf{R}{ij})^{-1} NN(\mathbf{x}')$. The equivariance is due to the fact that $\mathbf{x}'$ is invariant to rotations.
>
> Let us see what happens if $\mathbf{R}{ij}$ is not well defined, as we have estabilshed in the first round of this rebuttal. Then we in fact have that $\tilde{\mathbf{R}}^{-1} \mathbf{R}{ij}$ would be valid for any $\tilde{\mathbf{R}}$ that is a rotation around the z-axis. Then effectively any $\mathbf{D}(\tilde{\mathbf{R}}^{-1})\mathbf{x}'$ could have been obtained. The $NN$ is then no longer receiving an invariant input and now
> $$\mathbf{D}(\mathbf{R}{ij})^{-1} \mathbf{D}(\tilde{\mathbf{R}}) NN(\mathbf{D}(\tilde{\mathbf{R}}^{-1}) \mathbf{x}') \mapsto \mathbf{D}(\mathbf{R}) \mathbf{D}(\mathbf{R}{ij})^{-1} \mathbf{D}(\tilde{\mathbf{R}}) NN(\mathbf{D}(\tilde{\mathbf{R}}^{-1})\mathbf{x}') \, ,$$
> and since the $NN$ is not equivariant the Wigner-D matrix will not compensate for the arbitrary rotation $\tilde{\mathbf{R}}$.
>
> Thus when stating "replacing with a MLP will break the equivariance", it is not due to mere use of the MLP, since as just shown above this is actually no problem, but it only becomes a problem when a well-defined rotation matrix $\mathbf{R}_{ij}$ is lacking. [**A**] *The paper is still a bit suggestive in it's presentation as it gives the impression that the use of the MLP breaks equivariance, but it is rather the free rotation $\tilde{\mathbf{R}}$*.

---

> > ### Comment · Reviewer_rs3C · 2023-11-17
> >
> > [**B**] *Then it is still unclear how equation (10) overcomes this issue.* It says "To overcome these shortcomings", which refer to the randomness due to $\tilde{\mathbf{R}}$ I suppose, because that is the only valid mentioned shortcoming so far. Below equation 11 it is now mentioned in the updated version that the splitting addresses two issues: (a) mitigate randomness one MLP uses the unaltered input $NN(\mathbf{x}$; and (b) the other uses $\mathbf{x}''=\mathbf{D}(\mathbf{R}_{ij}) \mathbf{x}_i - \mathbf{x}_i$ "enabling MLP to extract direction information embedded in Wigner-D matrix". Regarding (a) it should be noted that this part is indeed not random, but it purely breaks equivariance eitherway since the $NN$ are not equivariant. Regarding (b) also this part is not equivariant because of the randomness due to $\tilde{\mathbb{R}}$, as discussed above. The equivariance would be reduced to only the randomness of the free rotation if $\mathbf{x}'$ instead of $\mathbf{x}''$ was used, however, the method used $\mathbf{x}''$ which contains the non-equivariant representation $\mathbf{x}$ as input. Thus neither of the parts in (11) are equivariant, but the first one is almost if $\mathbf{x}'$ would have been used.
> >
> > Thus, [**B**] *It is still unclear to me why $\mathbf{x}''$ is used*. I understand that equation (10) is a reparametrization of the original tensor product, and the identity holds due to linearity of the tensor product. But once the TPs are replaced by MLPs it does not make any sense to me why this split is useful. In fact, I would argue that you shouldn't do this and rather try to split the expression into an non-equivariant and equivariant part. I hope you could still be explained.
> >
> > [**C**] *Perhaps a specific question could be asked: Why not use $\mathbf{x}'$ instead of $\mathbf{x}''$ for the first MLP?*
> >
> > In conclusion, [**D**] *the main layers (11,12) are not equivariant, and it is in my opinion not appropriate to call it approximately equivariant since there is no part in it that is actually equivariant*. Just replacing something which was original equivariant by something is not equivariant at all does not make it approximately equivariant. If there is still a notion of approximate equivariance left, could you please explain it? Again, I may be mistaken here, but I am still not convinced of the appxorimate equivariance narrative. Please let me know if I'm overlooking something still.

---

> ### Author Response · Authors · 2023-11-18
> **Responses to reviewer**
>
> Thank you for your prompt reply. We are delighted to discuss equivariance with you. We update paper again based on your review. Please forgive our lengthy discussion, we hope that all concerns can be addressed well.
>
> # Response to statement [A]
> Thanks for pointing this out. As you say, although we mentioned randomness in the paper, we did not elaborate on its impact on equivariance. We have updated  expression in the paper. In short, **our modification clarifies that the destroy of equivariance arises from both ill-defined rotation and the non-equivariance of the MLP.** As you noted in the last formula, ill-defined transformation $\mathbf{D}(\tilde{\mathbf{R}})$ lead a term  $\mathbf{D}(\tilde{\mathbf{R}}) NN(\mathbf{D}(\tilde{\mathbf{R}}^{-1}) \mathbf{x}')$, and the non-equivariance of MLP prevents $\mathbf{D}(\tilde{\mathbf{R}}^{-1}) NN(\mathbf{x}') = NN(\mathbf{D}(\tilde{\mathbf{R}}^{-1}) \mathbf{x}')$. This ultimately leads to the breakdown of equivariance.
>
> # Responses to statements [B] and [C]
> **Let us assume that through training, MLP can learn perfect equivariance**, that is, $NN(\mathbf{D}(\mathbf{R})\mathbf{x}) = \mathbf{D}(\mathbf{R}) NN(\mathbf{x})$. We will discuss whether MLP can learn equivariance in Responses to statements [D].
>
> For your statement [C]. **If we replace $\mathbf{x}''$ with $\mathbf{x}'$ in Equation 11, the equality of Equation 11 does not hold, and the right hand side is not invariant.** In this case, the right side is expressed as $\mathbf{D}^{-1}(\mathbf{R} _{ij}) NN (\mathbf{x}') + \mathbf{D}^{-1}(\mathbf{R} _{ij}) NN(\mathbf{x}) = NN(\mathbf{x}) + \mathbf{D}^{-1}(\mathbf{R} _{ij} ) NN(\mathbf{x})$. The first term is equivariant but the second term is not.
>
> We clarify that **Equation 10 is a identity transformation of Equation 9 and the original CG tensor product**. Tensor product satisfies $(A + B) \otimes C = A \otimes C + B \otimes C$, so we can decompose Equation 9. We noticed that you mentioned "equation (10) is a reparametrization of the original tensor product, and the identity holds due to linearity of the tensor product". If what you want to express is the same as our understanding, please forgive our repeated explanation. Based on this fact, we cannot replace $\mathbf{x}''$ with $\mathbf{x}'$ for the first MLP in Equation 10.
>
> For statement [B]. In ideal case, $NN(\cdot)$ is strictly equivariant, and two conditions will be satisfied. First, $NN((\mathbf{D} - \mathbf{I}) (\mathbf{R})\mathbf{x}) = (\mathbf{D}- \mathbf{I})(\mathbf{ R}) NN(\mathbf{x})$. Second, $\mathbf{x}''$ is still an equivariant representation when $\mathbf{x}$ is an equivariant representation. This is because addition operation is equivariant. These two conditions ensure that ill-defined $\mathbf{R}_{ij}$ can be offset in the equation 11.
>
> In reality, it is difficult for MLP to learn perfect equivariance. For the analysis of this imperfect equivariance in equation 11, you can refer to response to statement [D].
>
> We appreciate your suggestion "split the expression into an non-equivariant and equivariant part". **In fact, we attempted to substitute one of the terms in equation 11 with a CG product.** However, any changes resulted in a degradation of performance. You can observe the outcomes of our previous experiments in the table below. Here, $\mathbf{x}$ denotes the replacement of $NN(\mathbf{x}{i})$ with $\mathbf{x}{i} \otimes \mathbf{S}(\mathbf{C})$. We employed a small HDNN (K=6, L=4) on the IS2RE dataset (MAE on ID task).
>
> | $NN(\mathbf{x} _{i})$ | $NN(\mathbf{x}'' _{i})$ | No replacing |
> |  ----  | ----  | ----  |
> | 622 | 614 | 591 |

---

> > ### Comment · Reviewer_rs3C · 2023-11-20
> >
> > Thank you for the added details. One last question regarding equation 11. If you say
> > $$\mathbf{x}i \otimes \mathbf{S}^L(\mathbf{r}{ij}) \approx \mathbf{D}^{-1}(\mathbf{R}{ij})(NN(\mathbf{x}i'') + NN(\mathbf{x}i))$$
> > I understand that simply replacing the $\mathbf{x}_i''$ with $\mathbf{x}_i'$ does not make sense, but what about approximating via
> > $$\mathbf{x}i \otimes \mathbf{S}^L(\mathbf{r}{ij}) \approx \mathbf{D}^{-1}(\mathbf{R}{ij})NN(\mathbf{x}i') \, ,$$
> > i.e., approximationg equation 9 instead of 10? It would give a much better starting point for the NN as it only has to learn how to handle the randomness in R, but is otherwise already automatically equivariant. Did you try this?

---

> ### Author Response · Authors · 2023-11-18
> **Responses to reviewer**
>
> # Response to statement [D]
> The function form of last term (METP) in equation 12 is strictly equivariant. We first discuss whether MLP can learn equivariance, and secondly, we discuss the performance of the last term (METP) in an approximately equivariant system.
>
> MLP is able to approximate continuous functions, including all equivariant functions (refer to figure 1 in [1]). Let's consider a simple case where we let MLP learn the length of a vector. **If training set contains all SO(3) transformations of vectors, it is possible for MLP to successfully learn the function for calculating the length.** In real, the size of the training set is very important. The more SO(3) transformation types the MLP receives, the more likely the MLP is to learn the correct equivariance.
>
> Equation 11 expands the SO(3) transformation through another method. Note that the MLP remains the same in the equation. The desired output is achieved only when the MLP exhibits equivariance for both $\mathbf{D}-\mathbf{I}$ and $\mathbf{I}$ transformations. To measure equivariance, we introduce an error term. Let $\mathbf{\lambda}$ be a constant vector. For all transformations and a given MLP, there exists $ \mathbf{\lambda} > | NN(\mathbf{D} \mathbf{x}) - \mathbf{D}NN(\mathbf{x}) \|$.
> Considering Equation 11 and the version without decomposition ($\mathbf{D}^{-1}(\mathbf{R} _{ij}) NN(\mathbf{x}')$), when the input undergoes a random transformation, the upper limit of error for the version without decomposition is $\mathbf{D}^{-1}(\mathbf{R} _{ij}) \mathbf{\lambda}$. Using the L2 Norm provides a stable upper limit, denoted as $e _{1} = | \mathbf{D}^{-1}(\mathbf{R} _{ij}) \mathbf{\lambda} | = | \mathbf{\lambda} |$. However, the upper limit of Equation 11 is $e _{2} = | 2 * \mathbf{\lambda}|$. The training process aims to reduce the upper limit of error through the loss function between output and label. If equation 11 and the version without decomposition achieve the same upper limit ($e _{1} = e _{2}$), the equivariant error of MLP needs to satisfy ($|\mathbf{\lambda} _{1}| = 2 | \mathbf{\lambda} _{2} |$). **In summary, the decomposition in equation 11 reduces the equivariant error of each MLP after training. When $\mathbf{\lambda} _{2}$ is small enough, we can refer to it as approximately equivariant.** Our MAD in Table 4 demonstrates the equivariance error of the overall model.
>
> **The essence of METP is CG tensor product. It remains strictly equivariant. You may worry that $\mathbf{x}$ is approximate spherical harmonics basis based. However, the equivariance on low-degree representations will be influence Slightly.** For example, the CG tensor product on type-1 vectors is equivalent to the cross product of 3D vectors. No matter what value the type-1 vector takes. The output of CG tensor product is strictly equivariant. Additioanlly, combination with attention coefficients is an equivariant process, since they are invariant.
>
> **For more rigorous expression, we have updated the explanation of equation 11. Furthermore, we also indicate that approximately equivariance requires an efficient training.** Please check section 3.2 in updated paper.
>
> Finally, we provide our macro-level expectations for MLP. In our equation, MLP replaces the function CG tensor product with a certain degree. But in fact, what it approximates may be a more complex function, such as a higher-order tensor product. Some operations can promote effective learning of MLP. For example, it is also possible to learn CG tensor product using $NN(\mathbf{x} _{i}, \vec{\mathbf{r}} _{ij})$. However, transforming to the z-axis can lead to a more efficient learning, because it can transform the tensor product into a simple matrix multiplication. Similarly, we expect that the MLP in Equation 11 can learn some equivariant information in $\vec{\mathbf{r}} _{ij}$. This information may be useful for embedding angular information (such as bond angles, dihedral angles). We consider that separated output is more conducive to parsing out direction information.
>
> [1] On the Universality of Rotation Equivariant Point Cloud Networks. Nadav Dym, Haggai Maron. ICLR 2021

---

> > ### Comment · Reviewer_rs3C · 2023-11-20
> >
> > Thank you also for clarifying the approximation accuracy. Firstly, I appologize for overlooking the METP term, which is clearly equivariant and which shows that the layer has both an equivariant and non-equivariant term.
> >
> > Regarding the approximation however, I see that the split gives a better approximation bound, in the case of equal equivariance of the NN in both scenarios. A last question I have on this point is the following. What incentive is there for the NN to be equivariant? It is only the training objective, right? (which is supposed to be invariant). This means that getting better equivariance properties probably means better performance. And thus one could conclude that under equal performance of the $\mathbf{x}'$ or two term decomposition, the latter will have a better equivariance bound.
> >
> > Do I understand this correctly; that the NN's will be more precise in their equivariance approximation because now there are effectively 2 terms pushing the NN to be equivariant, rather than just 1 term?

---

> ### Author Response · Authors · 2023-11-20
> **Responses to reviewer**
>
> Thanks again for your prompt reply.
>
> # Response to the comment "Thank you for the added details. One last question regarding ..."
> Yes, we have tried using equation 9, as evidenced by our ablation experiments ($NN(\mathbf{x}'')$ in Table 5). The experiment involved replacing $NN(\mathbf{x}'') + NN(\mathbf{x})$ with $NN(\mathbf{x}')$ and showed $NN(\mathbf{x}'') + NN(\mathbf{x})$ method achieves better result. Noticeable gaps were observed in our previous experiments, particularly on small datasets (like qm9).
>
> It is worth noting that **learning how to deal with randomness is essentially learning about equivariance.** Since only equivariance ensures that the ill-defined $\tilde{\mathbf{D}}$ inside the brackets can compensate for the $\tilde{\mathbf{D}}^{-1}$ outside the brackets, thereby eliminating randomness (like what you say in your second review). In either case, whether it is $NN(\mathbf{x}')$ or $NN(\mathbf{x}'') + NN(\mathbf{x})$, equivariance of $NN(\cdot)$ is common learning goal. From this perspective, starting from Equation 9 does not increase the learning facilitation.

---

> ### Author Response · Authors · 2023-11-20
> **Responses to reviewer**
>
> I am sorry that we made some errors when submitting our responses to openreview, which may have resulted in multiple emails you receive. We have responded to your two new concerns. You can check in the webpage.
>
> # Response to the comment "Thank you also for clarifying the approximation accuracy. Firstly ..."
> **Your understanding is correct. Equivariance is just a learning goal, and the process of approaching equivariance must be data-driven.** As I mentioned in the second round of reponse to your statement [D], if the training set contains all possible SO(3) transformations, the output of the neural network is likely to satisfy $NN(\mathbf{R}x)=\mathbf{R}NN(x)$. This is based on the knowledge that continuous functions contain equivariant functions.
>
> We now have two MLP terms, but they share weights, representing the same function. Under the equal performance of overall model, the equivariant error of MLP structure will be better if we use $\mathbf{x}''$ and $\mathbf{x}$. However, in our experiments, the training process tends to make the equal equivariance bound of individual MLP, thereby making the equivariance bound of the overall model (using $\mathbf{x}''$ and $\mathbf{x}$) better.
>
> Let’s briefly discuss the benefits of introducing MLP. MLP brings an advantage and a disadvantage. The advantage is that MLP can fit more complex equivariant functions compared to low-degree CG tensor products. The disadvantage is that its equivariance is data-driven, which needs sufficient molecular data. **In engineering, this disadvantage may be alleviated by SO(3)-transformation augmentation of the training set. Satisfactory models is possible in a variety of molecular tasks.** In our paper, we do not introduce SO(3) augmentation as it leads to an unfair comparison.

---

> > ### Comment · Reviewer_rs3C · 2023-11-22
> >
> > I raised my score further from 5 to 6.
> >
> > In contrast to what I initially assumed after reading the first version of the manuscript, the method is very well thought through. It is in good condition to be accepted. What could have further improved the paper is a more detailed/intuitive discussion on what mechanisms drive the neural network to be equivariant. The revisions made during the rebutal phase in my opinion are sufficient but still it feels like the insights presented during the discussion period could have broader implications, these are not really explicitly discussed in the paper, but could convey an interesting take home message. Overall, I believe the paper contains substantial contributions.

---

> > > ### Author Response · Authors · 2023-11-23
> > > **Updating Key Insights from the Discussion into the Manuscript**
> > >
> > > Thank you for acknowledging our work and for your suggestion. **We have recently released an updated version that includes additional detailed insights in the discussion phase. Due to space constraints, we have appended these detailed contents to Appendix B.2 and made corresponding references in the main text.** In future revisions, we plan to reorganize these significant insights into the main body of the manuscript.

---

> > > > ### Comment · Reviewer_rs3C · 2023-11-23
> > > >
> > > > Thanks a lot for your commitment in making this an excellent submission. I had my doubts at first, based on a misunderstanding and initial formulations which did not seem to make sense to me. These concerns have all been addressed, and I think the current submission is clear about its contributions. The paper is both theoretically sound, the experimental results are great, and the overall narrative is now good to follow. As such, my current opinion of the paper is clear accept (8). I updated my initial score accordingly.

---

### Author Response · Authors · 2023-11-22
**To Reviewers**

Dear Reviewers,

Thank you for your time and the positive evaluation of our manuscript. We have submitted the revised version, where we believe we have effectively addressed all of your concerns. We have highlighted the modified sections in red. Specifically, in the updated manuscript, we have: (a) underscored the role of randomness in $\mathbf{R} _{ij}$ and provided an explanation of its connection to non-equivariance; (b) introduced the role of the MLP in Equations 11 and 12 and explained the concept of approximate equivariance; (c) improved the analysis of OOD tasks of IS2RE in the experimental section; and (d) supplemented the implementation details of $\mathbf{R} _{ij}$ and included complexity analysis in the appendix.

As the discussion session is drawing to a close, we would like to confirm whether you have any remaining concerns. We sincerely appreciate your diligent efforts. Thank you once again.

### To Reviewer rs3C
Thanks for your constructive discussion, which significantly contributed to the enhancement of our manuscript. We have addressed your last questions. Details can be found in our latest responses. In short, 1. Our ablation experiments in Table 5 demonstrate that the structure $NN(\mathbf{x}'') + NN(\mathbf{x})$ outperforms the structure $NN(\mathbf{x}')$. 2. Regarding the equivariance of MLP, your understanding is accurate. Moreover, our design in Equation 11 and 12 aims to minimize the equivariant loss induced by MLP during training.

[Update after reading your last comment. We appreciate your suggestions and incorporate the detailed insights from the discussion into the manuscript. Thank you.]

### To Reviewer Liwe
Thanks for your review. We have addressed all of your concerns (one weekness and two questions). In responses, we bolded key sentences to enhance the clarity and facilitate understanding. Regarding the weakness you pointed out, we want to clarify that we did not claim the state-of-the-art performance on all the tasks. However, our method achieves the best average score in OC20 IS2RE, which is our primary focus.

### To Reviewer yQB9
Thanks you for your acknowledgment of our work

Best regards, Authors

---

### Meta-Review · Area_Chair_WnHT · 2023-12-09

**Metareview:**

This paper studies molecular representation learning with a hybrid approach combining equivariant modules with learnable ones. This work receives positive supports from reviewers and thus an accept is recommended.

**Justification For Why Not Higher Score:**

Some of the experimental results are not very strong.

**Justification For Why Not Lower Score:**

This work receives positive supports from reviewers.

---

### Decision · Program_Chairs · 2024-01-16

Accept (spotlight)